# Asymmetric ON-OFF processing of visual motion cancels variability induced by the structure of natural scenes

Juyue Chen[1], Holly B Mandel[2], James E Fitzgerald[3†]*, Damon A Clark[1,2,4,5†]*

[1]Interdepartmental Neuroscience Program, Yale University, New Haven, United States; [2]Department of Molecular, Cellular and Developmental Biology, Yale University, New Haven, United States; [3]Janelia Research Campus, Howard Hughes Medical Institute, Ashburn, United States; [4]Department of Physics, Yale University, New Haven, United States; [5]Department of Neuroscience, Yale University, New Haven, United States

**Abstract** Animals detect motion using a variety of visual cues that reflect regularities in the natural world. Experiments in animals across phyla have shown that motion percepts incorporate both pairwise and triplet spatiotemporal correlations that could theoretically benefit motion computation. However, it remains unclear how visual systems assemble these cues to build accurate motion estimates. Here, we used systematic behavioral measurements of fruit fly motion perception to show how flies combine local pairwise and triplet correlations to reduce variability in motion estimates across natural scenes. By generating synthetic images with statistics controlled by maximum entropy distributions, we show that the triplet correlations are useful only when images have light-dark asymmetries that mimic natural ones. This suggests that asymmetric ON-OFF processing is tuned to the particular statistics of natural scenes. Since all animals encounter the world's light-dark asymmetries, many visual systems are likely to use asymmetric ON-OFF processing to improve motion estimation.

**\*For correspondence:**
fitzgeraldj@janelia.hhmi.org (JEF);
damon.clark@yale.edu (DAC)

†These authors contributed equally to this work

**Competing interests:** The authors declare that no competing interests exist.

## Introduction

For any visual system, motion estimation is an important but computationally challenging task. To accurately extract motion signals from complex natural inputs, visual systems should take advantage of all useful information. One source of information lies in the stable statistics of the visual input, that is, in the regularities of natural scenes (*Geisler, 2008*). A strong version of this hypothesis is that visual systems are tuned, through evolution and experience, to the statistics of natural environments (*Chichilnisky and Kalmar, 2002*; *Olshausen and Field, 1996*; *Simoncelli and Olshausen, 2001*; *Srinivasan et al., 1982*). However, it remains unclear how visual systems use the statistics of natural scenes and the motion signals in them to aid in motion estimation (*Salisbury and Palmer, 2016*; *Sinha et al., 2018*).

Motion computation has long been understood algorithmically as selective responses to specific spatiotemporal correlations (*Fitzgerald et al., 2011*; *Poggio and Reichardt, 1973*; *Potters and Bialek, 1994*). For example, canonical models propose that animals extract motion from visual signals by detecting *pairwise* spatiotemporal correlations (*Adelson and Bergen, 1985*; *Hassenstein and Reichardt, 1956*). Higher order correlations could also contribute to motion computation, and Bayes optimal visual motion estimators can be written as a sum of terms specialized for detecting different correlation types (*Potters and Bialek, 1994*; *Fitzgerald et al., 2011*). This mathematical result follows from a Volterra series expansion, which provides a general and systematic way to represent nonlinear computational systems. Higher order correlations are also empirically relevant. For

example, triplet spatiotemporal correlations are the next lowest-order terms after pairwise correlations, and both humans and flies perceive motion in 'glider' stimuli that isolate triplet spatiotemporal correlations (*Clark et al., 2014*; *Hu and Victor, 2010*). The sensitivity to triplet spatiotemporal correlations shows that motion perception incorporates cues neglected by canonical motion detectors.

Interestingly, perceptual sensitivities to triplet spatiotemporal correlations prove that visual systems must consider the polarity of contrast when computing motion. This is because triplet correlations flip sign when contrast polarities are inverted, which means that the perceptual contribution of triplet correlations to a motion estimator reverses when all input contrasts are inverted. This contrast-polarity dependent motion processing has been hypothesized to be an adaptation to natural scenes, especially to the light-dark asymmetry of the contrast distribution (*Clark et al., 2014*; *Fitzgerald and Clark, 2015*; *Fitzgerald et al., 2011*; *Leonhardt et al., 2016*; *Nitzany and Victor, 2014*). For example, simulated motion detectors that were optimized to estimate motion in natural scenes exhibited contrast-polarity-dependent responses similar to flies (*Fitzgerald and Clark, 2015*; *Leonhardt et al., 2016*). These modeling studies suggest that contrast-polarity-dependent responses emerge from performance optimization in natural scenes, but they do not show that real visual systems use their sensitivity to triplet spatiotemporal correlations to improve motion estimates. This limitation arises because previous experimental studies measured sensitivities to only a few triplet correlations (*Clark et al., 2014*; *Leonhardt et al., 2016*). However, one cannot assess the utility of individual correlations in isolation (*Clark et al., 2014*; *Nitzany et al., 2016*), and naturalistic visual signals contain many spatiotemporal correlations with diverse spatiotemporal structures. Moreover, although previous analyses recognized that some kind of light-dark asymmetry is required for triplet correlation sensitivity to emerge in optimized motion estimators (*Clark et al., 2014*; *Fitzgerald and Clark, 2015*; *Fitzgerald et al., 2011*; *Leonhardt et al., 2016*), they did not discover which statistical regularities within natural scenes were sufficient for the observed motion signals to improve accuracy.

Here, we filled these gaps by systematically measuring the nonlinearities in *Drosophila* visual motion detection and relating them to light-dark asymmetries in natural scenes. We first systematically characterized low-order components of the fly's motion computation algorithm by modeling its visually evoked turning behavior with a Volterra series expansion (*Clark et al., 2011*; *Clark et al., 2014*; *Fitzgerald et al., 2011*; *Marmarelis and McCann, 1973*; *Poggio and Reichardt, 1973*; *Salazar-Gatzimas et al., 2016*). Through this framework, we extended canonical pairwise (second-order) motion computation models by adding a triplet (third-order) component that accounts for contrast-polarity-dependent motion computation. We evaluated the performance of the inferred algorithm across an ensemble of moving natural images and discovered that the third-order component improves velocity estimates by canceling image-induced variability in the second-order component. Finally, we leveraged maximum entropy distributions to develop a method for generating synthetic images with precisely controlled contrast statistics. This method revealed that the skewness of natural images allows the fly's sensitivity to triplet spatiotemporal correlations to improve its canonical motion estimates.

## Results

### The structure of natural scenes induces variability in second-order motion estimates

To evaluate how canonical motion detectors performed with natural scene inputs, we simulated responses of the Hassenstein-Reichardt Correlator (HRC) to rigidly translating natural scenes. The HRC exemplifies canonical motion detectors, which rely exclusively on pairwise spatiotemporal correlations to estimate motion (*Adelson and Bergen, 1985*; *Hassenstein and Reichardt, 1956*) (*Figure 1A*). It can be equivalently written as a motion energy model (*Adelson and Bergen, 1985*). We used a database of natural, panoramic photographs to create naturalistic motion stimuli (*Meyer et al., 2014*). In particular, we first converted the photographs' luminance signals into local contrast signals (*Figure 1B*, *Figure 1—figure supplement 1*). We then rigidly translated these natural images at various horizontal velocities to simulate full-field motion signals (*Badwan et al., 2019*; *Dror et al., 2001*; *Fitzgerald and Clark, 2015*; *Leonhardt et al., 2016*). This rigid translation of images mimics the motion produced by an animal's pure rotation, during which visual objects all

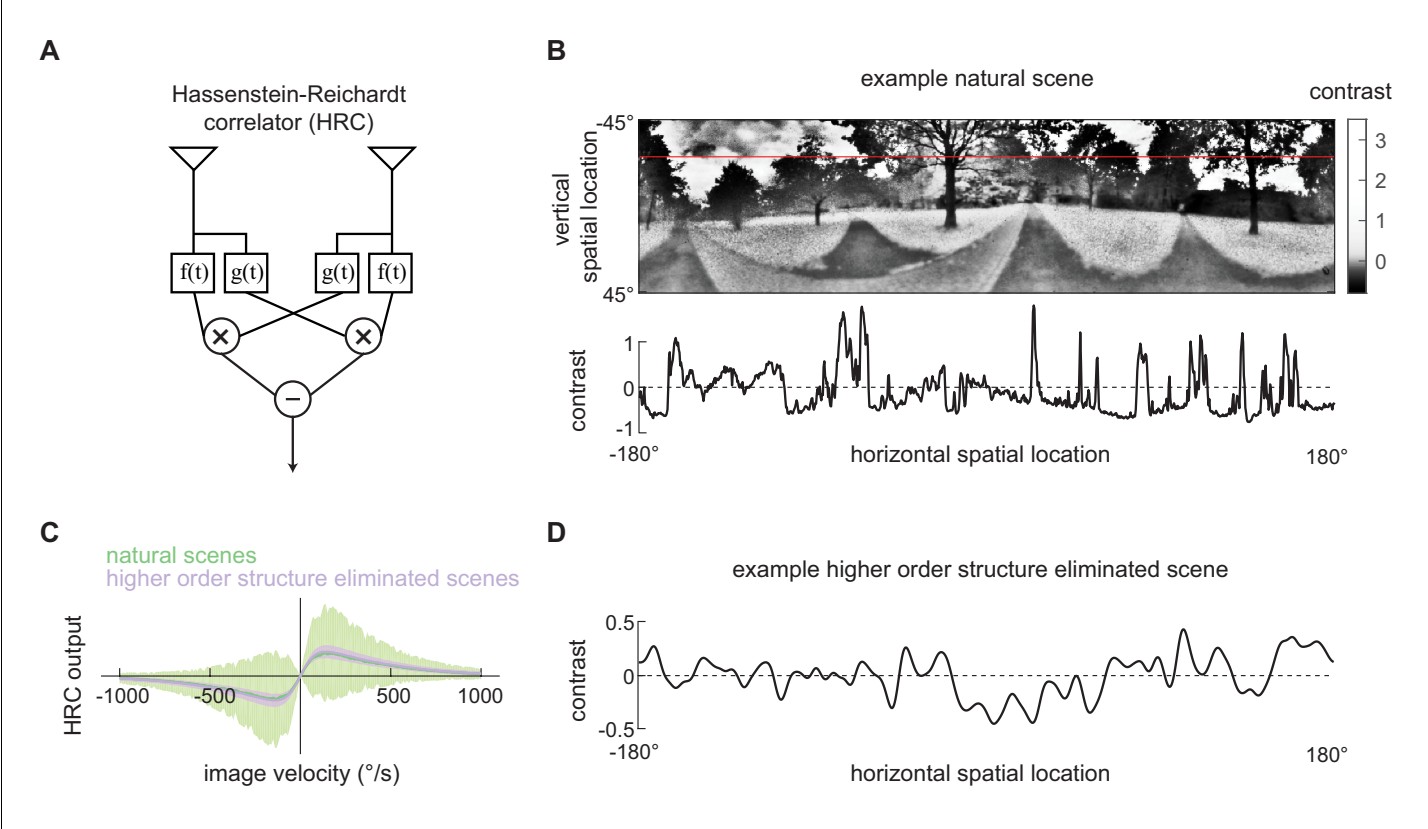

**Figure 1.** Second-order motion detectors perform poorly with natural scene inputs. (**A**) Schematics of the Hassenstein-Reichardt correlator (HRC). Each half of the HRC receives inputs from two nearby points in space, which are filtered in time with filters $f(t)$ and $g(t)$, and then multiplied together. The full HRC receives outputs from two symmetric halves with opposite direction tuning and subtracts two outputs. (**B**) An example two-dimensional photograph from a natural scene dataset (*top*), including a one-dimensional section (image) through the photograph (*bottom*), indicated by the red line. So that the image can be viewed clearly, the contrasts in the photograph were mapped onto gray levels so that an equal number of pixels were represented by each gray level. (**C**) Average response (*line*) and variance (*shaded*) of the outputs of an HRC (equivalent to a motion energy model; *Adelson and Bergen, 1985*) when presented with naturalistic motion at various velocities. Images were sampled from natural scenes (*green*) or from a synthetic image dataset in which all higher order structure was eliminated (*purple*, see Materials and methods). (**D**) Example synthetic image in which all higher order structure was eliminated.

The online version of this article includes the following figure supplement(s) for figure 1:

**Figure supplement 1.** Converting luminance signals into contrast signals (see Materials and methods).

move at the same rotational velocity and occlusion does not change over time. Real motion through an environment generates more complex signals than this, but rigid translations are straightforward to compute and rotational visual stimuli are known to induce the rotational optomotor response that we focus on in this manuscript.

The spatiotemporal contrast signals from these (image, velocity) pairs were used as inputs to the HRC model, and we evaluated the model's output for fixed image velocities across different scenes (*Figure 1C*, Materials and methods). The model generated a mean response that was linearly tuned for small velocities, peaked at around 130 °/s, and then decayed to zero for fast speeds (*Figure 1C green line*). However, we observed substantial variance about the mean response, and this variance implies that different natural scenes generated different second-order motion estimates, even when moving at the same velocity (*Figure 1C green shading*). This is consistent with the finding that canonical second-order motion detectors generate variable responses with natural scene inputs (*Dror et al., 2001*; *Fitzgerald and Clark, 2015*; *Sinha et al., 2018*).

Next we sought to investigate how the higher order structure of natural scenes influences the performance of the second-order motion estimates. Though canonical motion detectors use only pairwise spatiotemporal correlations, higher order statistics of static images, such as contrast

kurtosis, influence the detector's variance (*Clark et al., 2014*; *Fitzgerald and Clark, 2015*). To demonstrate this, we generated a synthetic image set in which we preserved the second-order statistics of natural scenes, including their spatial correlation function and contrast variance, but eliminated all higher order structure (*Figure 1D*, Materials and methods). When the higher order structure was eliminated, the HRC's average tuning was unchanged, but there was a marked decrease in the variance (*Figure 1C purple*). This demonstrates that higher order structure in natural scenes induces variability in canonical motion estimates.

## Modeling fly motion computation with second- and third-order Volterra kernels

To investigate how real visual systems compute motion, we wanted to systematically characterize an animal's motion computation at the algorithmic level (*Marr and Poggio, 1976*). Motion computation requires a nonlinear transformation to form a motion estimate from the visual stimulus (*Borst and Egelhaaf, 1989*; *Fitzgerald et al., 2011*; *Poggio and Reichardt, 1973*). We approximated this nonlinear transformation using a Volterra series expansion (*Marmarelis and McCann, 1973*; *Marmarelis, 2004*; *Schetzen, 1980*; *Wiener, 1966*). Similar to the Taylor series from calculus, the Volterra series is a polynomial description of a nonlinearity, with a first-order kernel that describes linear transformations, a second-order kernel that captures quadratic terms, and higher-order kernels that combine to represent a wide variety of nonlinearities beyond the second-order. However, many polynomial terms can be needed to describe some nonlinearities. For instance, the polynomial description of a compressive, saturating nonlinearity is inefficient, and it can be easier to describe such transformations using alternative nonlinear model architectures, such as linear-nonlinear cascade models (*Dayan and Abbott, 2001*). We emphasize that the Volterra kernel description is explicitly algorithmic, as it aims to summarize the overall system processing without considering the mechanisms leading to this processing.

Volterra kernels are useful for studying visual motion processing because they allow us to rigorously group response properties by their order (*Fitzgerald et al., 2011*; *Potters and Bialek, 1994*), thereby permitting us to clearly describe both canonical and contrast polarity-dependent components of the behavior. For example, the second-order kernel is equivalent to the canonical motion detecting algorithms, as it explains the sensitivity to pairwise spatiotemporal correlations (*Fitzgerald et al., 2011*; *Salazar-Gatzimas et al., 2016*). Second-order Volterra kernels, along with related spike-triggered covariance methods (*Bialek and van Steveninck, 2005*; *Sandler and Marmarelis, 2015*; *Schwartz et al., 2006*), have been used to model second-order behavior and neural processing in flies and primates (*Clark et al., 2011*; *Marmarelis and McCann, 1973*; *Poggio and Reichardt, 1973*; *Rust et al., 2005*; *Salazar-Gatzimas et al., 2016*). However, the second-order kernel cannot capture the system's sensitivity to triplet spatiotemporal correlations. We therefore minimally extended the depth of the Volterra series expansion to include the third-order kernel. The third-order kernel directly measures sensitivities to triplet spatiotemporal correlations and probes ON/OFF asymmetries in motion processing.

## Experimental measurements of Volterra kernels in fly behavior

We focused on how the fly responds to correlations between nearest-neighbor pixels in the visual input, which corresponded roughly to a single ommatidium separation (*Buchner, 1976*) (*Figure 2A*). The second-order kernel describes how the behavioral response is influenced by the product of contrasts at each pair of spatiotemporal points in the visual input (*Figure 2B blue*). In comparison, the third-order kernel describes how the response is influenced by the product of contrasts at each triplet of spatiotemporal points in the visual input (*Figure 2B green*). Note that triplet spatiotemporal correlations could in principle be computed across three distinct spatial locations, but our analysis focused on triplet spatiotemporal correlations distributed across two nearest-neighbor pixels.

In order to extract Volterra kernels, especially higher-order ones, we needed a large amount of data. We thus developed a high-throughput setup to measure turning in walking flies in response to visual stimuli (*Figure 2C*) (*Creamer et al., 2018*; *Creamer et al., 2019*). In this setup, a fly's optomotor turning response serves as a readout of its motion perception (*Götz and Wenking, 1973*; *Hassenstein and Reichardt, 1956*), which allowed us to characterize the fly's motion computation algorithm by measuring its visually-evoked turning response. Flies spend a large portion of their lives

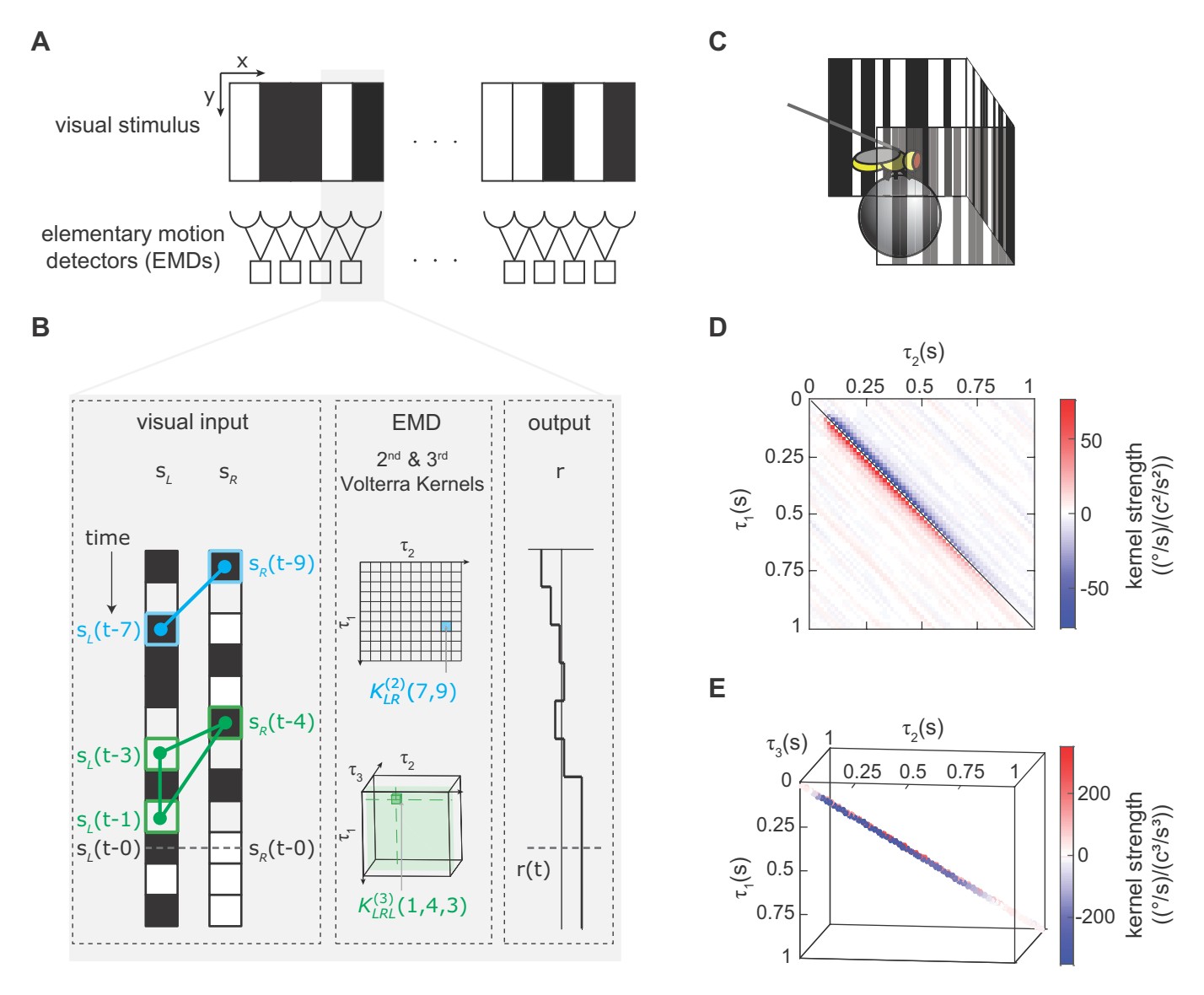

**Figure 2.** We modeled the fly's motion computation algorithm with second- and third-order Volterra kernels and extracted the kernels using reverse-correlation (Appendix 1, Materials and methods). (**A**) Visual depiction of our model, where the fly's motion computation system consists of a spatial array of elementary motion detectors (EMDs), and each EMD receives inputs from two neighboring spatial locations. We presented flies with vertically uniform stimuli with 5°-wide pixels, roughly matching ommatidium spacing. (**B**) Diagram showing how the output of one EMD at time $t$, indicated by gray dashed line, is influenced by the second- and third-order products in the stimulus. *Left*: visual inputs of one EMD, $s_L$ and $s_R$. The visual stimulus contained products of pairwise and triplet points with various spatiotemporal structure. One specific pairwise product is highlighted (*blue barbell*) and one specific triplet product is highlighted (*green triangle*). *Middle*: The motion computation of the EMD is approximated by the second-order kernel (*blue*) and the third-order kernel (*green*). The second-order kernel (*blue*) $K_{LR}^{(2)}(\tau_1, \tau_2)$ is a two-dimensional matrix. For example, the response at time $t$ is influenced by the products of $s_L(t-7)$ and $s_R(t-9)$ with weighting $K_{LR}^{(2)}(7, 9)$. The third-order kernel (*green*) $K_{LRL}^{(3)}(\tau_1, \tau_2, \tau_3)$ is a three-dimensional tensor. The response at time $t$ is influenced by $s_L(t-1)s_R(t-4)s_L(t-3)$ with weighting $K_{LRL}^{(3)}(1, 4, 3)$. *Right*: turning response at time $t$ is influenced by all pairwise and triplet products in the visual stimulus, with weightings given by the second- and third-order kernel elements. (**C**) Diagram of the fly-on-a-ball rig. We tethered a fly above a freely-rotating ball, which acted as a two-dimensional treadmill. We presented stochastic binary stimuli, and measured fly turning responses. (**D**) The extracted second-order kernel. The color represents the magnitude of the kernel, with red indicating rightward turning and blue indicating leftward turning to positive pairwise spatiotemporal correlations. Above the diagonal line, the matrix represented left-tilted pairwise products (example in B) and below the diagonal line represents right tilted pairwise products. (**E**) The extracted third-order kernel. For visualization purposes, we show only the two diagonals with the largest magnitude.

The online version of this article includes the following figure supplement(s) for figure 2:

*Figure 2 continued on next page*

*Figure 2 continued*

**Figure supplement 1.** Using reverse-correlation to extract second- and third-order kernels from the measured turning response to stochastic binary stimulus (Materials and methods, Appendix 1).

standing and walking on surfaces, making walking optomotor responses ethologically critical (*Carey et al., 2006*).

We extracted the second- and third-order Volterra-kernels with reverse-correlation methods. To do this, we presented flies with spatiotemporally uncorrelated binary stimuli on a panoramic screen around the fly, measured their turning responses, and correlated the behavior at each time to the stimuli preceding it (Materials and methods, Appendix 1, *Figure 2—figure supplement 1*) (*Clark et al., 2011*; *Mano and Clark, 2017*; *Salazar-Gatzimas et al., 2016*). The measured second-order kernel showed positive and negative lobes (*Figure 2D*). The positive lobe below the diagonal indicates that flies turned to the right when presented with positive correlations in the rightward direction. This second-order kernel is consistent with classical models of motion computation and with previous neural and behavioral measurements (*Clark et al., 2011*; *Marmarelis and McCann, 1973*; *Salazar-Gatzimas et al., 2016*). The measured third-order kernel also showed both positive and negative values (*Figure 2E*), and we will dissect its detailed structure later in this manuscript. However, we first set out to evaluate how the third-order kernel contributed to motion estimation across an ensemble of moving natural images.

## The third-order kernel improves velocity estimation for moving natural scenes

The kernels were fit to turning behavior, so the output of the model to moving visual stimuli is the predicted optomotor turning response. Following previous work (*Clark et al., 2014*; *Fitzgerald and Clark, 2015*; *Leonhardt et al., 2016*; *Poggio and Reichardt, 1973*; *Potters and Bialek, 1994*; *Sinha et al., 2018*), we hypothesized that optomotor turning responses provide a proxy for the fly's velocity estimate. Using the fitted behavioral model, we could thus investigate how accurately the fly's velocity estimate tracks the true image velocity. We evaluated the fly's motion computation performance with a simple and specific metric: when an entire natural image translates rigidly with constant velocity, how accurately does the behavioral algorithm predict the image velocity (*Figure 3A*)? Specifically, does the fly use its sensitivity to triplet spatiotemporal correlations to improve velocity estimation?

We sampled the velocities from a zero-mean Gaussian distribution with a standard deviation of 114 °/s: this distribution roughly matched turning distributions in walking flies (*DeAngelis et al., 2019*; *Katsov and Clandinin, 2008*). Crucially, because we measured the Volterra kernels, we could separate the fly's predicted output into two components: the canonical second-order response, $r^{(2)}$, and the non-canonical third-order response, $r^{(3)}$ (*Figure 3A*). The second-order response is the output from the second-order kernel, and it describes how the fly responded to naturalistic second-order spatiotemporal correlations in the stimulus. Similarly, $r^{(3)}$ is the output from the third-order kernel, and it describes how the fly responded to naturalistic triplet spatiotemporal correlations. This separation allowed us to ask how the pairwise and triplet correlations are individually and jointly used to estimate motion.

We quantified how well the model's responses predicted the image velocity using the Pearson correlation coefficient (*Clark et al., 2014*; *Fitzgerald and Clark, 2015*; *Leonhardt et al., 2016*). This metric supposes that the model response and image velocity are linearly related, and its value summarizes intuitively the mean-squared-error of the best linear fit between the model's output and the image velocity. When the correlation coefficient has an absolute value near 1, the model closely tracks image velocity, while a value near 0 indicates no linear relationship between model and image velocity. The responses derived from the second-order kernel, $r^{(2)}$, correlated positively with the true velocity (*Figure 3B blue*), indicating that the second-order response matches the behavioral direction (*Clark et al., 2011*; *Hassenstein and Reichardt, 1956*; *Salazar-Gatzimas et al., 2016*). Interestingly, the isolated third-order response, $r^{(3)}$, anti-correlated with true image velocities (*Figure 3B green*). This means that the fly's third-order response on its own would predict that the fly turns in the direction opposite to the presented motion. However, when $r^{(3)}$ was added to $r^{(2)}$, the accuracy

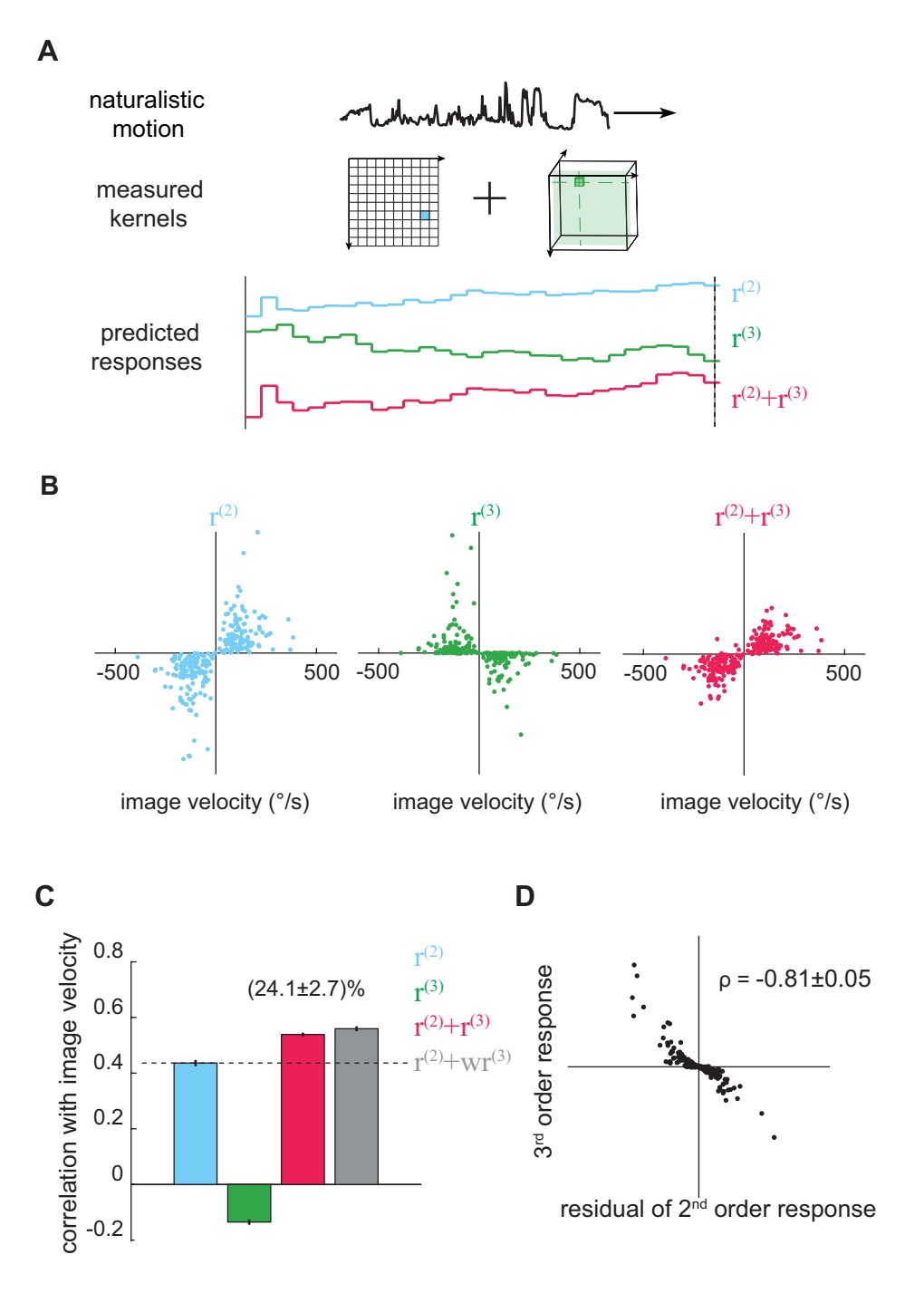

**Figure 3.** The third-order kernel improved motion estimation in natural scenes. (**A**) Predicting responses of the second- and third-order kernels to rigidly moving scenes. *Top*: natural scenes rigidly translating with constant velocities. *Middle*: cartoon of the second- and third-order kernels. *Bottom*: second-order response (*blue*), third-order response (*green*), the predicted motion estimate (*red*) is the summation of $r^{(2)}$ and $r^{(3)}$. (**B**) Scatter plot of $r^{(2)}$, $r^{(3)}$ and $r^{(2)} + r^{(3)}$ against image velocity over the ensemble of moving images. 10,000 independent trials were simulated, and 1000 trials were plotted here. (**C**) Pearson correlation coefficients between responses of each kernel and the true image velocities ($\rho$ = 0.44 ± 0.01, -0.14 ± 0.01, 0.54 ± 0.01, 0.56 ± 0.01, from *left* to *right*; $w$ = 1.39 ± 0.01; mean ± SEM across 10 groups of 1000 trials). (**D**) Scatter plot between $r^{(3)}$ and the residual in $r^{(2)}$, computed by subtracting a scaled image velocity from $r^{(2)}$ (Materials and methods). $\rho$ represents the Pearson correlation coefficient mean ± SEM across 10 groups (Materials and methods).

The online version of this article includes the following figure supplement(s) for figure 3:

*Figure 3 continued on next page*

*Figure 3 continued*

**Figure supplement 1.** The improvement added by the third-order kernel persists across a wide range of velocities.

**Figure supplement 2.** The length scale of local mean luminance computation affected the performance of the measured kernels.

of the full motion estimator increased by ~25% compared to $r^{(2)}$ alone (*Figure 3B red*, *Figure 3C*). This important result shows that the third-order responses improve velocity estimates only in conjunction with second-order responses.

To understand this counterintuitive finding, it's useful to recognize that the second-order response is influenced by both the image velocity and the structure of the natural scene. For example, recall that the output of the HRC depended both on the velocity of motion and on the particular image that was moving (*Figure 1C*). Thus, one way to improve the accuracy of the response is to reduce scene-dependent variability in the second-order estimate. To investigate whether this interpretation explained the observed improvement, we calculated the residuals of the second-order responses by subtracting the best linear fit of the image velocity and plotted them against the third-order responses. We found that the third-order signal was strongly anticorrelated with this scene-induced residual in the second-order response (*Figure 3D*). This means that the fly's sensitivity to triplet spatiotemporal correlations indeed canceled scene-dependent variability in the second-order motion estimator to improve the accuracy of motion estimation across natural scenes.

Since the magnitude of the second-order kernel and third-order kernel were each measured experimentally, our model combined $r^{(2)}$ and $r^{(3)}$ with a 1:1 ratio. Nevertheless, we were interested in whether the fly could have done better with alternate weighting coefficients, so we fit a linear regression model to reweight $r^{(2)}$ and $r^{(3)}$ to best predict image velocity. Strikingly, we found the optimized relative weighting between $r^{(2)}$ and $r^{(3)}$ was near one, and the performance of the best weighted model was only marginally better than the empirical model (*Figure 3C gray*). Thus, the measured second- and third-order kernels were weighted near optimally for performance in naturalistic motion estimation.

We also wanted to understand how the improvement added by $r^{(3)}$ depended on the parameters of our simulation. To see how it depended on the width of the image velocity distribution, we varied the standard deviation over an order of magnitude. The improvement did not depend strongly on the variance of the velocity (*Figure 3—figure supplement 1*). We also asked how the contrast computation affected the performance of the measured algorithm. When we previously converted luminance into contrast signals, we computed local contrasts on a length scale of 25° (measured by full-width-at-half-maximum), because that is the approximate spatial scale of surround inhibition measured in flies (*Arenz et al., 2017*; *Freifeld et al., 2013*; *Srinivasan et al., 1982*). When we swept this spatial scale from 10° to 75°, the improvement added by the third-order kernel first increased, peaked at around 30°, and then decreased to negative values after 40° (*Figure 3—figure supplement 2A–E*). When we computed the contrast over time, instead of space, we observed improvements on timescales less than 100 ms, comparable to measured timescales involved in early visual neurons that compute temporal derivatives (*Behnia et al., 2014*; *Srinivasan et al., 1982*; *Yang et al., 2016*) (*Figure 3—figure supplement 2FG*). However, the third-order term hurt performance when contrasts were computed on longer timescales. These results show that contrast computations influence the utility of the measured third-order kernel, with maximal utility occurring in a regime that approximately matches the contrast computation of the fly eye.

## Visualizing the measured third-order kernel with impulse responses

Since the measured third-order Volterra kernel improved motion estimates, we wanted to characterize it in more detail. To better visualize the third-order kernel, we rearranged its elements in an impulse response format (*Figure 4AB*, Materials and methods). The impulse response of a system is its output when presented with a small and brief input, called an impulse. This impulse may consist of a change in contrast at a single point, in which case the impulse response captures the linear response of the system. Analogously, if the impulse consists of a contrast triplet over three points in space and time, then the triplet impulse response captures the system's response to the interactions of those three points, after accounting for those responses already explained by linear or second-order impulse responses.

Triplet correlation impulse responses are useful because they allow one to rapidly digest how different triplet correlations will affect behavior. For example, in *Figure 4A*, we colored three occurrences of triplets that have the same spatiotemporal structure and their corresponding triplet impulse responses. We set the origin of the impulse responses to be the most recent point in the triplet, because the system could not respond to the interaction of three points before all three points were presented. Since the spatiotemporal structures of these three triplets are the same, the three impulse responses have the same shape. Note that a negative impulse, consisting of an odd number of dark elements within the triplet, would drive turning in the opposite direction. In *Figure 4B*, we represented impulse responses of different triplet with colormaps and used ball-stick cartoons to show the relative temporal distances between the points in each triplet. The predicted time course of the behavioral effect is easy to discern, and the kernel predicts that the behavioral consequences of triplet correlations will last almost a second. We can more compactly understand the relative magnitudes of the behavioral effects by summing the impulse responses over time (*Figure 4C*) (*Salazar-Gatzimas et al., 2016*). As expected, the impact of different triplet correlations varies significantly in both direction and magnitude.

## Verification of the third-order kernel measurement

We verified the reliability of our third-order kernel measurement in two ways. First, we tested the statistical significance of the measured kernel directly. We extracted an ensemble of null kernels by applying the reverse-correlation analysis to the measured behavioral responses and temporally-shifted visual stimuli (Materials and methods). By comparing summed kernel elements in the empirical and null kernels, we found that many terms in the third-order kernel were statistically significant at the p=0.05 level (*Figure 4C*). Significance was especially common when the temporal distance between the points in the triplet spatiotemporal correlation was less than 0.1 s.

Second, we measured the fly's sensitivity to triplet spatiotemporal correlations with third-order glider stimuli (*Figure 4D*, *Figure 4—figure supplement 1*). Third-order glider stimuli are binary stimuli that lack pairwise correlations and are enriched in specific triplet spatiotemporal correlations (*Clark et al., 2014*; *Hu and Victor, 2010*). We used the measured third-order kernel to predict responses to the glider stimuli. Most of the measured responses were quantitatively predicted by the third-order kernel (*Figure 4D*). Several gliders elicited smaller behavioral responses compared to the kernel prediction; such differences might be attributable to induced long-range spatial correlations in glider stimuli (*Clark et al., 2014*; *Hu and Victor, 2010*), which are not captured by our measured nearest-neighbor kernel. Nevertheless, the successes revealed by this independent experimental test strongly suggest that we had enough statistical power to reliably fit the third-order kernel to the behavioral data.

## The second- and third-order kernels share temporal structure

Multiple models propose that sensitivity to pairwise and triplet spatiotemporal correlations could emerge simultaneously from the same nonlinear step in the fly brain (*Fitzgerald and Clark, 2015*; *Leong et al., 2016*; *Leonhardt et al., 2016*). We were thus curious whether the measured second- and third-order kernels had a common temporal structure. To compare the second-order and third-order kernels, we simplified the third-order kernel to a two-dimensional approximation (*Figure 4—figure supplement 2ABC*), rearranged the second-order kernel into the impulse response format (*Figure 4—figure supplement 2D*), and computed summed kernel strengths to obtain one-dimensional representations for both kernels (*Figure 4E*). We compared the second- and third-order kernel elements at the same temporal offsets (*Figure 4E top*). In the case of pairwise correlations, the temporal offset was determined by the temporal distance between the left and the right points, and in the case of triplet correlations, the temporal offset was determined by the average temporal distance between the left and right points. The summed kernel strengths showed that the second-order and third-order kernels had similar sensitivities to temporal delays between the input pixels, with peak sensitivity at the shortest delays in our experiment (*Figure 4E bottom*). An analysis employing the singular value decomposition yielded similar results, and also showed comparable kinetics in the behavioral responses to pairwise and triplet correlations (*Figure 4—figure supplement 2EFG*). These similarities suggest that the second- and third-order responses originate in common physiological processes.

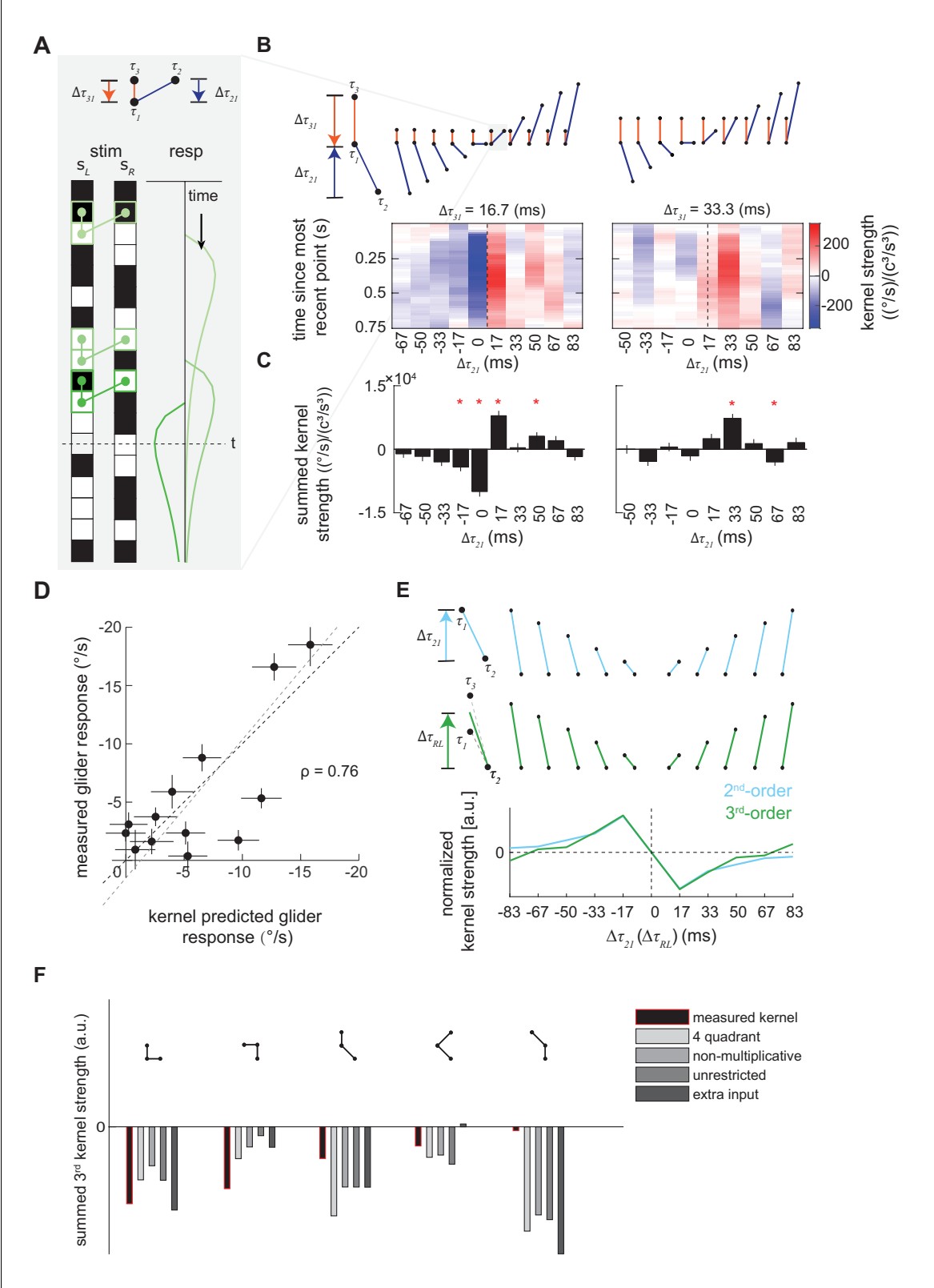

**Figure 4.** Characterization and validation of the measured third-order kernel. (**A**) Triplet impulse response description. *Top*: the ball-stick diagram represents the relative spatiotemporal position of three points in a triplet. The red line denotes the temporal distance between the two left points, $\Delta\tau_{31}$, and the blue line denotes the temporal distance between the more recent point on the left and the sole right point, $\Delta\tau_{21}$. *Bottom*: Three specific example occurrences of the triplet elicit three impulse responses. The response at time $t$ is the sum of the impulse responses to all previous

*Figure 4 continued on next page*

*Figure 4 continued*

occurrences of the triplet. The first triplet (*lightest green*) involves two black points and one white point, so their product is positive, and it elicits an impulse response with positive sign. The triplet occurs far from current time *t*, so its influence on the current response is small. The last triplet (*darkest green*) involves two white points and one black point, so the product is negative and it elicits an impulse response with flipped sign. It is close to current time *t*, and has a large influence on the current response. (B) Third-order kernel visualized using an impulse-response format (Materials and methods). *Top*: the ball-stick diagrams as in (A). *Bottom*: the color map plots the 'impulse response' to the corresponding triplets, and color represents the strength of the kernel. Different panels represent different $\Delta\tau_{31}$. In each color map, $\Delta\tau_{31}$ is fixed, the columns represent $\Delta\tau_{21}$, and the rows represent the time since the most recent point in each triplet. The dashed lines indicate the place where the right point is in the middle of the two left points in time. (C) The summed strength of the third-order kernel along each column in A. Error bars represent SEM calculated across flies (n = 72), and significance was tested against the null kernel distribution (*p<0.05, two-tailed z-test (Materials and methods)). (D) The scatter plot between the measured responses to third-order glider stimuli (Materials and methods, *Figure 4—figure supplement 1*) against responses predicted by the third-order kernel (Materials and methods). The correlation between the predicted and measured responses is 0.76. Black dashed line is unity; gray dashed line is the best linear fit. (E) The measured second- and third-order kernel share temporal structures. *Top*: the ball-stick diagrams represent the relative spatiotemporal positions of the two points in each pair (*blue*), and three points in each triplet (*green*). *Bottom*: The kernel strength of the second-order kernel (*blue*) and third-order kernel (*green*) summed across all elements sharing the same spatiotemporal structures, that is summed over rows in *Figure 4—figure supplement 2CD* (Materials and methods). *Figure 4—figure supplement 3*, (F) The extracted third-order kernels from four optimized motion detectors (*Fitzgerald and Clark, 2015*) compared to the measured kernel from the fly. The summed kernel strength is summed across all elements which shared the same spatiotemporal structures diagramed above.

The online version of this article includes the following figure supplement(s) for figure 4:

**Figure supplement 1.** Flies turned in response to third-order spatiotemporal correlations presented in binary glider stimuli.
**Figure supplement 2.** Rearranging the second- and third-order kernels and computing singular value decompositions on the rearranged kernels.
**Figure supplement 3.** Four models optimized to estimate image velocities in natural scenes, adapted from *Fitzgerald and Clark (2015)*.

## Comparing the measured third-order kernel to optimal motion estimators

A recent theoretical study proposed several motion detectors whose parameters were optimized for velocity estimation in natural scenes (*Fitzgerald and Clark, 2015*) (*Figure 4—figure supplement 3*). In order to compare our measured third-order kernel to those of these optimized motion detectors, we presented stochastic binary stimuli to these detectors and extracted their third-order kernels using reverse-correlation. We found that the third-order kernels of the optimized models were usually similar to each other (*Figure 4F*), which is consistent with prior analyses (*Fitzgerald and Clark, 2015*). The measured third-order kernel consistently agreed with the optimized kernels in its signs, and in some cases, the kernels were also similar in magnitude. However, certain kernel elements differed markedly between the optimized models and the behaviorally measured kernel. Perhaps most noticeably, the behavioral kernel was much smaller than the optimized kernels for correlations whose spatiotemporal structure involved large delays between the points (*Figure 4F*, third and fifth kernel elements). Such differences between the optimized models and the measured behavior could indicate suboptimalities in the fly brain. However, they could also result from unrealistic constraints imposed on the model optimization, such as fixed temporal processing and restricted model structures (*Fitzgerald and Clark, 2015*; *Leonhardt et al., 2016*). The measured kernel therefore provides valuable new data to inform theoretical work assessing the optimality of biological motion estimators.

## Positive skewness is sufficient for the third-order kernel to improve motion estimates

Which features of natural images allow the measured third-order kernel to improve motion estimates? The natural scene dataset is comprised of heterogeneous individual images (*Figure 5A*), so we calculated the contrast mean, variance, skewness, and kurtosis of each image individually. The variance describes the scale of the contrast variation; the skewness quantifies imbalance between contrasts above and below the mean; and the kurtosis roughly characterizes the frequency of extreme bright and dark points. Each of these statistics showed a wide distribution over the image ensemble (*Figure 5—figure supplement 1ABCD*). These statistics were also highly dependent on each other: a positively skewed image often had high variance and was highly kurtotic (*Figure 5B*, *Figure 5—figure supplement 1E*). These strong relationships make it difficult to isolate the effects of individual statistics within the image ensemble.

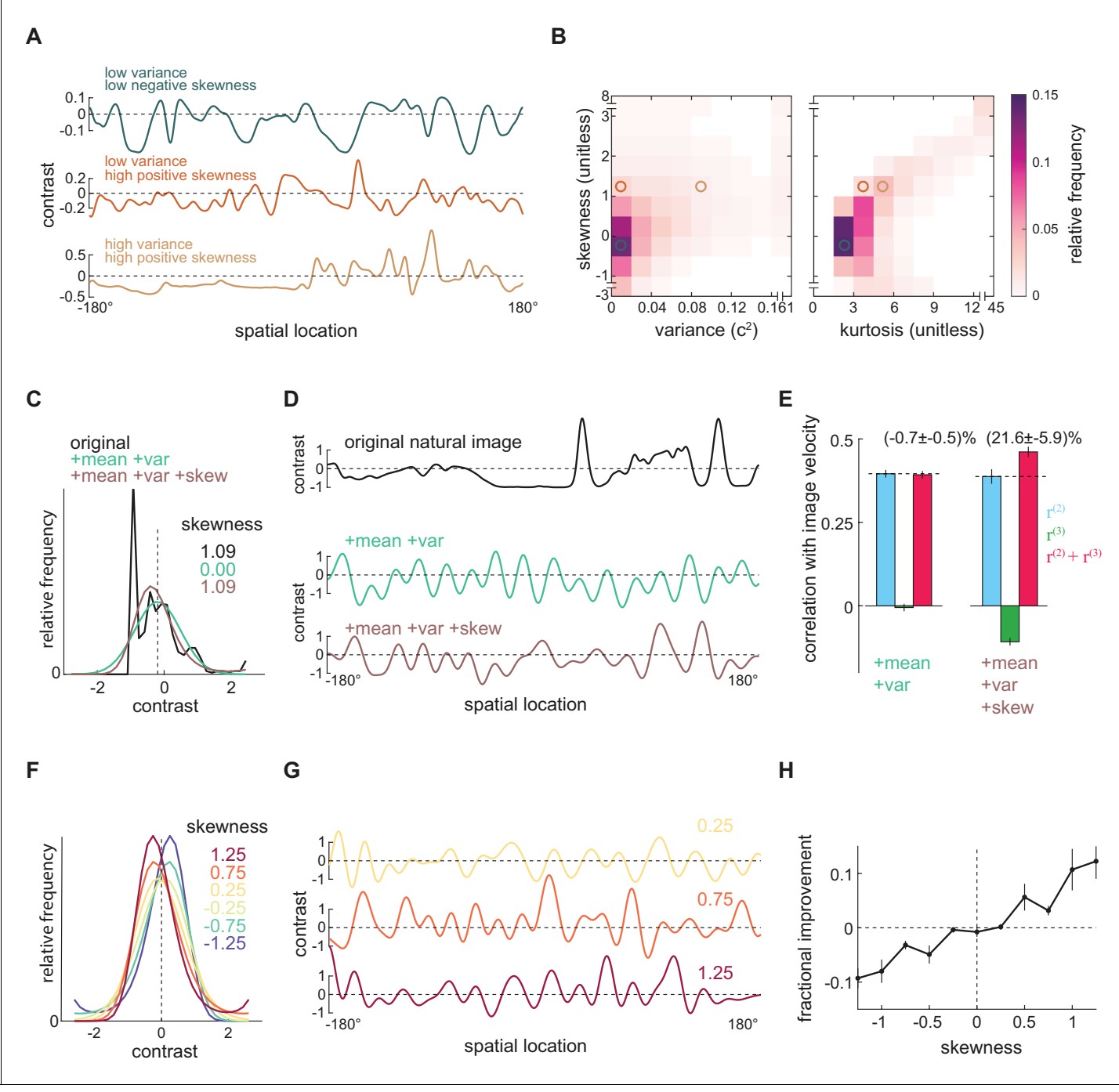

**Figure 5.** Positive skewness is sufficient for the third-order kernel to improve motion estimation. (A) Three example natural scenes with different degrees of variance and skewness. (B) Joint-density maps of individual image statistics over the ensemble of natural images, showing the relationship between skewness and variance (*left*) and skewness and kurtosis (*right*). (C) Contrast distributions of example images from the natural scene dataset and two synthetic image datasets. The natural image is shown (*black*), along with a maximum entropy distribution (MED) with matched mean and variance, denoted by +mean +var (*green*) and an MED with matched mean, variance, and skewness, denoted by +mean +var +skew (*brown*). (D) Example images from the natural scene dataset and two synthetic image datasets corresponding to three contrast distributions in (C). (E) The Pearson correlation coefficient between true image velocities and each kernel's responses in the two synthetic datasets +mean +var (*green*) and +mean +var +skew (*brown*) (F) Example of MEDs in six synthetic datasets, in which the image skewness ranged from −1.25 to 1.25. (G) Example of synthetic images in three synthetic datasets, corresponding to MEDs in (F) with constrained skewness of 0.25 (*top*), 0.75 (*middle*), and 1.25 (*bottom*). (H) Improvement added by the third-order response as a function of synthetic image skewness.

The online version of this article includes the following figure supplement(s) for figure 5:

*Figure 5 continued on next page*

*Figure 5 continued*

**Figure supplement 1.** Natural scenes have heterogenous contrast statistics.

**Figure supplement 2.** Using maximum entropy distributions (MEDs) to generate synthetic images with controlled image statistics.

**Figure supplement 3.** The performance of the measured kernels in various synthetic image datasets.

To isolate the effects of individual statistics, we therefore generated several different synthetic image datasets that have alternate contrast statistics (Materials and methods). To generate each image, we constructed a synthetic contrast distribution and sampled pixel contrasts from this distribution (Appendix 2, *Figure 5—figure supplement 2A*). In this way, we could manipulate the statistics of the image by constraining various statistics of the distribution. In particular, we constrained the distribution to have specific lower order moments, such as mean, variance, and skewness. A distribution is not solely determined by its lower order moments, so there can be many distributions sharing the same lower order moments (*Figure 5—figure supplement 2A*). Among all such distributions, we chose the most random one, known as the maximum entropy distribution (MED) (*Berger et al., 1996*; *Jaynes, 1957*; *Schneidman et al., 2006*; *Victor and Conte, 2012*). Because we can specify these lower order moments independently, we can ask whether specific statistics are *sufficient* to generate the improvement added by the third-order signal.

We began by generating two synthetic image datasets (Materials and methods). In the first dataset, we generated a synthetic image for each natural image that had the same contrast mean and variance. To do this, we first found an MED whose contrast mean and variance matched those of the natural image (*Figure 5C*, *green*). We then generated a single synthetic image by sampling from this MED (*Figure 5D green*). In the second dataset, we required the synthetic image to have the same contrast mean, variance, and skewness as the original image (*Figure 5CD*, *brown*). By retaining the skewness, these synthetic images retained naturalistic light-dark asymmetries. We then asked how the third-order response affected velocity estimates across these two synthetic image datasets. When only the mean and variance of natural scenes were retained, the third-order response was near zero (*Figure 5—figure supplement 2A*, *green*) and did not improve motion estimation (*Figure 5E*). However, when the synthetic scenes were constrained to be naturalistically light-dark asymmetric, the improvement added by the third-order kernel was recovered (*Figure 5E*), with magnitude comparable to what was observed for the natural scene dataset (*Figure 3C*).

Finally, we wanted to see whether the degree of skewness controlled the magnitude of the improvement. We therefore generated synthetic image datasets in which we systematically varied the image skewness (*Figure 5FG*). In these synthetic images, the degree of skewness determined how much the third-order response could improve the full motion estimate (*Figure 5H*). When the images were negatively skewed, the third-order response correlated with image velocity (*Figure 5—figure supplement 3C*). However, since it also positively correlated with the residual in the second-order response, adding it to the second-order response decreased the model's overall performance (*Figure 5—figure supplement 3CD*). When the images were positively skewed, the third-order response became anticorrelated with the residual in the second-order response and thus improved the overall motion estimates (*Figure 5H*, *Figure 5—figure supplement 3CD*). These synthetic image sets show that positive image skewness is sufficient for the third-order signal to improve motion estimates. Thus, the measured algorithm for motion estimation leverages light-dark asymmetries found in natural scenes to improve motion estimates.

## Discussion

In this study, we first fit a Volterra series expansion to model the fly's turning behavior in response to binary stochastic stimuli, and both second- and third-order terms in the Volterra series contributed to the turning behavior. We then evaluated the model's output when it was presented with an ensemble of rigidly translating natural scenes. There, the second- and third-order terms of the model combined to produce outputs that better correlated with image velocities. There is no *a priori* reason to assume that a model fit to explain turning behavior would necessarily predict the image velocity. Therefore, these results can be taken together to motivate the hypothesis that the magnitude of the fly's turning response is determined by an internal estimate of velocity. Furthermore, this

estimate is specifically tailored for natural environments, since we found that the third-order kernel relies on light-dark asymmetries that are present in natural scenes but not in arbitrary images. Since skewed scenes are prevalent across natural environments, and many visual systems exhibit ON-OFF asymmetric visual processing (*Chichilnisky and Kalmar, 2002*; *Jin et al., 2011*; *Mazade et al., 2019*; *Pandarinath et al., 2010*; *Ratliff et al., 2010*; *Zaghloul et al., 2003*), many animals are likely to use similar strategies for motion perception.

## Direct demonstration that triplet correlations improve velocity estimation for natural images

The idea that features of the visual motion computation serve to improve performance in natural environments is conceptually appealing and theoretically powerful. For example, prior studies have found that optimized motion detectors had triplet correlation sensitivities similar to those measured from the fly visual system (*Fitzgerald and Clark, 2015*; *Leonhardt et al., 2016*). Although it is intriguing that biologically relevant response properties emerged in optimized motion detectors, the link from contrast-polarity dependent motion computation to naturalistic motion estimation remained indirect. Here we have provided the first direct demonstration that third-order components of the fly's motion computation algorithm improve velocity estimation for moving natural scenes. This direct demonstration had not been possible before because prior measurements were limited to a narrow range of correlations, and it was unclear how the measured cues interacted with unmeasured components of the motion estimation algorithm. For example, here we found that the third-order kernel appeared counterproductive when viewed in isolation, but the way flies incorporated triplet correlations was easily interpretable via the deficits of the second-order motion estimator. Remarkably, this problem could have persisted if motion computation needed to be understood in the context of additional visual motion cues that involve longer spatial-scales or higher order nonlinearities, which have been neglected in this study. This suggests that it might be sufficient to comprehensively characterize motion computation with a few local and low-order visual cues, which is encouraging for the approach outlined here. On the other hand, a mechanistically accurate model of visual motion processing could eventually summarize the relevant cues in a more succinct and less abstract way.

## Flies use triplet correlations to cancel scene-dependent variability in second-order cues

We found that the third-order responses in flies were anti-correlated with the natural image velocities (*Figure 3BC*). Nevertheless, they improved velocity estimation when added to the second-order responses (*Figure 3D*). This result appears counter-intuitive at first but can be understood. Spatiotemporal correlations are influenced by both the motion and local structure of the scene, but motion-driven behaviors should ignore fluctuations stemming from the scene's structure as much as possible. Pairwise and triplet spatiotemporal correlations are related to contrast variance and skewness, respectively, which are correlated across the ensemble of natural scenes (*Figure 5B*). This means that fluctuations in second-order signals tend to be accompanied by fluctuations in third-order signals. Therefore, with the right weighting, second-order and third-order signals may collaborate to reduce the image-induced signal fluctuations in the motion estimate (*Clark et al., 2014*). Indeed, we found that the third-order responses improved motion estimates because they helped to cancel variability in the second-order responses induced by the structure of natural scenes. This finding highlights a generally important but underappreciated point about cue combination and population coding in neural systems. Although a neuron's tuning is often used as a proxy for its involvement in stimulus processing, even untuned neurons can contribute productively to downstream decoding if their responses are correlated with noise in the tuned neuronal population (*Zylberberg, 2017*).

## Large computational benefits can underlie small behavioral effects

The HRC has explained a large number of behavioral phenomena and neural responses, and it is reasonable to ask how much we have gained by extending its second-order algorithmic description to a third-order one. The magnitude of behavior elicited by the third-order kernel is small compared to the second-order kernel's contribution. However, the magnitude of the behavioral effect and the

magnitude of the underlying performance gain can differ significantly. For example, the largest reported turning response elicited by a third-order glider stimulus is less than 20% of that elicited by second-order glider stimuli (*Clark et al., 2014*), yet the performance gain afforded by a motion detector designed to detect its defining third-order correlation exceeded 30% (*Fitzgerald and Clark, 2015*). Similarly, here we predict that many natural images would elicit little output from the third-order kernel (*Figure 3B*), yet third-order responses improved the correlation between the model output and velocity by ~25% (*Figure 3C*). These performance gains are modest, but they are comparable to the inarguable benefits provided by spatial averaging and could be ecologically relevant to the fly (*Dror et al., 2001*; *Salazar-Gatzimas et al., 2018*). Since we have only approximated the system with a spatially localized low order polynomial, we also expect that the improvements we observed here represent a lower bound on the total effects provided by the full mechanism underlying light-dark asymmetric motion processing. Indeed, longer range and higher order motion detection models can nearly double motion estimation accuracy while predicting realistic glider response magnitudes (*Fitzgerald and Clark, 2015*). It will be interesting to investigate whether mechanistically accurate models that explain the origin of the third-order kernel also reveal larger performance improvements.

## Asymmetric ON-OFF processing could affect motion processing across the animal kingdom

Here, we showed that flies systematically exploit contrast asymmetries in natural scenes to improve their visual motion estimates. This resonates with previous work showing that many visual systems process ON and OFF signals asymmetrically to improve other aspects of visual processing (*Kremkow et al., 2014*; *Mazade et al., 2019*; *Pandarinath et al., 2010*; *Ratliff et al., 2010*). Moreover, flies and vertebrates share striking anatomical and functional properties in their motion detection circuits (*Borst and Helmstaedter, 2015*; *Clark and Demb, 2016*; *Sanes and Zipursky, 2010*), likely because they are solving similar problems with similar constraints. We thus expect that many visual systems use ON-OFF asymmetric processing to improve visual motion perception. Nevertheless, it remains unclear how similar or different the details of such strategies will be across the animal kingdom. Indeed, although both primates and insects respond to third-order glider stimuli, their patterns of response differ (*Clark et al., 2014*; *Hu and Victor, 2010*; *Nitzany et al., 2017*). ON-OFF asymmetric visual processing also varies in other ways, and there is evidence that contrast adaptation in ON and OFF pathways is different between primate and salamander retinas (*Chander and Chichilnisky, 2001*).

Adaptations to differences in both habitat and early sensory processing could potentially explain these divergences. Here, we found positive and negative contrast skewness in different terrestrial scenes (*Figure 5—figure supplement 1C*), but positive skewness was most prevalent across the scenes. If certain habitats feature scenes that are predominantly negatively skewed, our work predicts that animals living in these habitats should have opposite third-order responses to flies (*Figure 5H*). More generally, natural scenes in different habitats are known to be similar in some statistics and different in others (*Balboa and Grzywacz, 2003*; *Burkhardt et al., 2006*). Interestingly, positive skewness might be particularly common in natural luminance distributions, because luminance signals are the product of many independent factors that generically combine to produce lognormal distributions (*Richards, 1982*). Nevertheless, the skewness level that matters for motion detecting circuits also depends on earlier processing operations in the eye (*Figure 3—figure supplement 2C*). In our numerical experiments, we computed local contrast signals. The spatial scales of this preprocessing influenced the statistics of the resulting contrast, which in turn influenced the performance of the motion computation. Biologically, this suggests that signal processing in early visual circuits can strongly influence how downstream circuits organize their computations (*Dror et al., 2001*; *Fitzgerald and Clark, 2015*). Alternatively, early sensory processing might be tailored to accommodate the computational requirements of downstream processing. These possibilities are not mutually exclusive, and in both cases, the early visual processing must work in concert with the downstream motion detectors to form robust and consistent perceptions.

## Volterra kernels systematically characterize nonlinear motion computations

The algorithm used by the visual system to extract motion signals is a nonlinear transformation from light detection to motion estimates. There are numerous ways to characterize a nonlinear system. In many cases, visual neuroscientists have purposefully designed stimuli, such as sinusoidal gratings, plaids, and gliders, to probe specific nonlinearities in the system (*Clark et al., 2014*; *Creamer et al., 2018*; *Euler et al., 2002*; *Fisher et al., 2015*; *Haag et al., 2016*; *Hu and Victor, 2010*; *Movshon et al., 1985*; *Rust et al., 2005*; *Salazar-Gatzimas et al., 2018*; *Salazar-Gatzimas et al., 2016*). In other cases, they have used stochastic stimuli to fit simple predictive models, such as linear-nonlinear models, generalized linear models, cascade models, and normalization models to capture a restricted but relatively broad set of biologically plausible nonlinearities (*Dayan and Abbott, 2001*; *Leong et al., 2016*; *Maheswaranathan et al., 2018*; *McIntosh et al., 2016*; *Salazar-Gatzimas et al., 2016*; *Simoncelli and Heeger, 1998*).

Here, we approximated the nonlinear system with Volterra kernels (*Marmarelis and Naka, 1972*; *Wiener, 1966*). This represents a general and systematic approach to nonlinear system identification, since (1) one need not make strong assumptions about the system to measure its kernels, (2) higher order kernels can in principle be added to characterize the system arbitrarily well, and (3) a complete set of kernels predicts the system's output for arbitrary input signals. One major limitation of this approach is that higher order kernels become progressively more difficult to fit as the number of kernel elements increases. This makes the approach most practical when a few low-order terms already capture conceptually important variables. Here, we leveraged the fact that second-order kernel capture the canonical models for visual motion estimation while third-order kernel probes ON/OFF asymmetries in motion processing. These two kernels can be related to distinct statistics of natural scenes.

Polynomial approximations to complex nonlinear systems have also been useful in other domains of neuroscience. For example, the experimental phenomenon of frequency-dependent long-term potentiation can be explained by extending canonical pairwise spike-timing-dependent plasticity models to include the relative timing of three spikes (*Pfister and Gerstner, 2006*; *Sjöström et al., 2001*). This makes learning sensitive to third-order correlations (*Gjorgjieva et al., 2011*). In the field of texture perception, researchers have long sought low order statistics that explain whether two patterns are texturally discriminable (*Julesz, 1962*; *Julesz et al., 1973*; *Julesz et al., 1978*). Similar to our findings for motion perception, both natural scene statistics and upstream visual processing play important roles (*Hermundstad et al., 2014*; *Portilla and Simoncelli, 2000*; *Tkacik et al., 2010*). As a final example, understanding how neural network structure impacts dynamics was aided by formally expanding the network's connectivity matrix into low-order connectivity motifs (*Hu et al., 2014*; *Trousdale et al., 2012*). These motifs might relate to measurable properties of the neocortex (*Song et al., 2005*).

## Velocity estimation is a useful approximation to motion computation

In this paper, we evaluated the fly's motion computation algorithm by measuring the accuracy of velocity estimation. Prior studies have often hypothesized that velocity estimation is a key requirement of motion processing and optomotor circuitry (*Clark et al., 2014*; *Dror et al., 2001*; *Fitzgerald and Clark, 2015*; *Fitzgerald et al., 2011*; *Poggio and Reichardt, 1973*; *Potters and Bialek, 1994*). It is thus reassuring that a model that better fit optomotor behavior also predicted image velocity more accurately. Nevertheless, motion computation is involved in perceptual tasks beyond the optomotor response, such as detecting looming stimuli (*Card and Dickinson, 2008*; *Zacarias et al., 2018*). In such tasks, the goal may not be to estimate the velocity of the visual object, but spatiotemporal correlations might nevertheless be useful (*Nitzany and Victor, 2014*). In addition, fly motion detecting neurons respond to static sinusoids or local luminance changes without obvious relevance for motion processing (*Fisher et al., 2015*; *Gruntman et al., 2018*; *Salazar-Gatzimas et al., 2018*), which suggests that motion computation algorithms might be jointly optimized alongside the detection of other visual features. Finally, visual systems have to coordinate with motor systems to achieve accurate sensorimotor transformations, so one should take the properties of the motor system into consideration when evaluating the performance of a motion computation algorithm (*Dickinson et al., 2000*). Future work could consider more sophisticated evaluation

metrics that better reflect the total ethological relevance of the visual environment to the fly. As we continue to distill the factors that combine to set the ultimate performance criteria, the use of velocity estimation is likely to remain a simple, useful, and insightful approximation.

## Potential mechanisms underlying the measured light-dark asymmetries

Visual systems in both vertebrates and invertebrates split into ON and OFF pathways that process light and dark signals separately and asymmetrically (*Balasubramanian and Sterling, 2009*; *Chichilnisky and Kalmar, 2002*; *Clark and Demb, 2016*; *Leonhardt et al., 2016*; *Ratliff et al., 2010*; *Ravi et al., 2018*; *Sagdullaev and McCall, 2005*; *Salazar-Gatzimas et al., 2018*). Some of these differences could result from biological constraints. Others could be an ethologically relevant adaptation to light-dark asymmetries found in the natural world. Either way, it is difficult to extrapolate from asymmetric neuronal processing of light and dark signals to functional asymmetries in downstream processing, including behavior. In this study, we used the behavioral turning responses to measure asymmetries in the flies' motion computation algorithm, instead of examining ON and OFF processing channels at the neuronal level. Since optomotor sensitivity to triplet spatiotemporal correlations is necessarily a functional consequence of underlying asymmetric visual signal processing, we could thus directly link light-dark asymmetries in natural scenes to the functional impact of ON-OFF asymmetric neural circuitry. It is similarly important to identify additional light-dark asymmetric behaviors that can clarify the functional role of other light-dark asymmetries in visual processing.

Having established the functional relevance of ON-OFF asymmetric visual processing, it is next important to find its neural implementation. Previous work has suggested that front-end nonlinearities could account for certain optomotor illusions in flies (*Bülthoff and Götz, 1979*), and it is conceivable that such nonlinearities could generate contrast asymmetric motion responses (*Clark and Demb, 2016*; *Fitzgerald et al., 2011*). However, several simple front-end nonlinearities can improve motion estimation without inducing the observed triplet correlation responses (*Fitzgerald and Clark, 2015*). Alternatively, nonlinear processing at the level of direction-selective T4 and T5 neurons could also generate the asymmetries we observed here. Indeed, differentially affecting T4 and T5 activity, either through direct silencing or by manipulating upstream neurons, alters the behavioral responses of flies to triplet correlations (*Clark et al., 2014*; *Leonhardt et al., 2016*), and parallel experiments in humans similarly find that contrast-asymmetric responses are mediated by neurons separately modulated by moving ON and OFF edges (*Clark et al., 2014*). Yet, it remains unclear whether asymmetric responses of T4 and T5 are inherited from upstream neurons. For instance, contrast adaptation could differ between the two pathways (*Chichilnisky and Kalmar, 2002*), and incompletely rectified inputs to T4 and T5 could generate asymmetrical responses to light and dark inputs (*Salazar-Gatzimas et al., 2018*). The weightings of T4 and T5 signals in downstream circuits could also result in contrast asymmetric phenomena. This rich landscape of possibilities motivates us to think that multiple mechanisms are likely to be involved. By measuring behavior and distilling the abstract algorithmic properties of the system, we will be able to constrain the contributions of individual circuit components without confining ourselves to an overly narrow class of mechanistic models.

## Relating algorithm and implementation in fly visual motion estimation

David Marr famously asserted that neural computation needs to be understood at both the algorithmic and implementational levels (*Marr and Poggio, 1976*). The benefits of this dual understanding go both ways. On one hand, the brain is immensely complicated, and an algorithmic theory can provide an invaluable lens for making sense of its details. On the other hand, the nuances of neuronal implementation can lead to new algorithmic questions and mechanistically satisfying answers. Marr used the optomotor response of flies to articulate his philosophy over forty years ago, and the community is still leveraging this problem to unravel the subtle relationships between algorithm and mechanism in the brain. The HRC model provided the first algorithmic theory of fly visual motion estimation, and this model's insights into the roles of spatial separation, differential time delays, and nonlinear signal integration have now been verified mechanistically (*Arenz et al., 2017*; *Fisher et al., 2015*; *Haag et al., 2016*; *Leong et al., 2016*; *Salazar-Gatzimas et al., 2016*; *Takemura et al., 2017*). They still provide the bedrock of our understanding. Yet the precise

mathematical form and mechanistic origin of the nonlinearity remain controversial, with different papers pointing out compelling roles for membrane voltages, intracellular calcium signals, and ON-OFF pathways (*Badwan et al., 2019*; *Gruntman et al., 2018*; *Haag et al., 2016*; *Leong et al., 2016*; *Leonhardt et al., 2016*; *Salazar-Gatzimas et al., 2018*; *Wienecke et al., 2018*). None of this complexity invalidates the core insights of the HRC, nor does the HRC's domain of success warrant apathy toward the fundamental importance of these unexpected findings. Instead of algorithm and mechanism providing parallel or hierarchical goals, they should be treated as parts of one integrated understanding of the circuit.

## Materials and methods

### Fly husbandry

Flies were grown at 20℃, 50% humidity in 12-hr day/night cycles on a dextrose-based food. Flies used for the behavioral experiment were non-virgin wildtype (*D. melanogaster*: WT: +; +; +) females between 24 and 72 hr old.

### Psychophysics

The fly's turning behavior was measured with the fly-on-a-ball rig, as described in previous studies (*Clark et al., 2011*; *Creamer et al., 2018*). The fly was tethered above a ball floating on a cushion of air. The ball served as a treadmill such that the fly could walk and turn while its position and orientation were fixed. The rotational response of the fly was the averaged rotation magnitude of the ball in 1/60s bins with an angular resolution of ~0.5°. Panoramic screens surrounded the fly, covering 270° horizontally and 106° vertically (*Creamer et al., 2019*). A Lightcrafter DLP (Texas Instruments, USA) projected visual stimulus to the screens with chrome green light (peak 520 nm and mean intensity of 100 cd/m$^2$). The spatial resolution of the projector was around 0.3° and the projector image was updated at 180 Hz. The rig's temperature was 34-36°.

### Visual stimuli

Visual stimuli varied along the horizontal axis in 5° pixels and were uniform along the vertical dimension. Since the panoramic screen was 270° wide, the horizontal axis was divided into $270/5 = 54$ pixels, so the screen was divided into 54 vertical bars.

We used two types of binary stochastic visual stimuli for kernel extraction, a three-bar-block stimulus type, and a four-bar-block stimulus type. In the 3 (4)-bar-block stimuli, each block contained 3 (4) neighboring vertical bars that flickered white or black independently in space and time. The identical blocks then repeated periodically around the fly. Since there are 54 bars, the entire visual field was divided into 18 (14.5) blocks. Each bar updated its contrast every 1/60 second.

We used third-order glider stimuli (*Hu and Victor, 2010*) to directly measure the fly's sensitivity to three-point correlations. Third-order glider stimuli are binary patterns of black and white pixels. In each glider stimulus, one can enforce a three-point spatiotemporal correlation. Here, we considered only three-point spatiotemporal correlations that involved two neighboring points in space. We described the specific configuration of each glider with a four-parameter scheme, $(\Delta\tau_{31}, \Delta\tau_{21}, L\backslash R, P)$. We defined the 1st point to be the more recent one of the two points sharing a spatial location, while the other point at this spatial location was defined to be the 3rd point. The final point, which was in a position adjacent to the 1st and 3rd points, was defined to be the 2nd point. The temporal interval between the 2nd point and 1st point was denoted as $\Delta\tau_{21}$, and $\Delta\tau_{31}$ was defined similarly. For example, $\Delta\tau_{31} = 1$ means that the 3rd point is 1 frame (16 ms) before the 1st point. Although $\Delta\tau_{31}$ is positive by definition, $\Delta\tau_{21}$ can be positive or negative. We used *L* and *R* to indicate whether the 1st and the 3rd points are on the left or right of the 2nd point. As detailed previously (*Clark et al., 2014*; *Hu and Victor, 2010*), in positive parity gliders ($P = +1$) one or three of these three points are white, whereas negative parity gliders ($P = -1$) have one or three of the points black. We illustrated the configuration of each glider using a 'ball-stick' diagram, where the x-axis represents space, the y-axis represents time, time runs downward, and the plus (minus) sign denotes the polarity of the glider (*Figure 4—figure supplement 1A*). Overall, we presented $52 = 13 \times 2 \times 2$ different stimuli: thirteen different temporal intervals, each with two directions and two polarities (*Table 1*).

**Table 1.** Statistics of responses to third-order glider stimuli with different spatiotemporal structures.

| Index | $\Delta\tau_{31}$ (16 ms) | $\Delta\tau_{21}$ (16 ms) | $(n_{P=1}, n_{P=-1})$ | $(p_{P=1}, p_{P=-1})$ |
|---|---|---|---|---|
| 1 | 1 | 0 | (18, 12) | (0.0003, <0.0001) |
| 2 | 2 | 0 | (35, 29) | (0.0299, 0.0003) |
| 3 | 3 | 0 | (14, 8) | (0.4218, 0.0092) |
| 4 | 4 | 0 | (14, 8) | (0.0201, 0.3552) |
| 5 | 1 | 1 | (18, 13) | (<0.0001, <0.0001) |
| 6 | 2 | 2 | (35, 30) | (0.0026, <0.0001) |
| 7 | 3 | 3 | (14, 9) | (0.2323, 0.0875) |
| 8 | 4 | 4 | (14, 9) | (0.6778, 0.1700) |
| 9 | 1 | -1 | (8, 8) | (<0.0001, 0.0044) |
| 10 | 2 | 1 | (8, 8) | (0.0041, 0.0617) |
| 11 | 1 | 2 | (8, 8) | (0.0044, 0.6713) |
| 12 | 3 | 1 | (21, 21) | (0.5396, 0.4470) |
| 13 | 3 | 2 | (21, 21) | (0.1042, 0.0203) |

## HRC model

We constructed a classical Hassenstein-Reichardt correlator (HRC) model (*Hassenstein and Reichardt, 1956*). The output $r_{\mathrm{HRC}}(t)$ was defined as

$$r_{\mathrm{HRC}}(t) = [s_1 * f_{\mathrm{HRC}}][s_2 * g_{\mathrm{HRC}}] - [s_2 * f_{\mathrm{HRC}}][s_1 * g_{\mathrm{HRC}}],$$

where $s_1(t)$ and $s_2(t)$ denote the contrast signals from two spatial locations, * denotes convolution in time, and $f_{\mathrm{HRC}}(t)$, $g_{\mathrm{HRC}}(t)$ are the temporal filters of the delay line and the non-delay line. In particular,

$$f_{\mathrm{HRC}}(t) = t \, \exp\left(-\frac{t}{\tau_{\mathrm{HRC}}}\right),$$

for $t \geq 0$, $f_{\mathrm{HRC}}(t) = 0$ for $t < 0$, and

$$g_{\mathrm{HRC}}(t) = \frac{d}{dt} f_{\mathrm{HRC}}(t),$$

where $\tau_{\mathrm{HRC}} = 20$ ms.

## Modeling the fly's motion computation algorithm with Volterra kernels

We approximated the fly's motion computation algorithm with second- and third-order Volterra kernels. We provide a detailed description of this model in *Section 1* of Appendix 1. In brief, we discretize space into pixels with $\Delta x = 5°$ resolution, discretize time into time bins with $\Delta t = 1/60$ s resolution, and index locations in space with an integer subscript $\xi$. We modeled the response of the fly $r(t)$ as the sum of an array of elementary motion detectors (EMDs) acting at each position in space:

$$r(t) = \sum_{\xi} r_{\xi}(t),$$

where $r_{\xi}(t)$ denotes the response of EMD at spatial location $\xi$. That term is itself the sum of the second-order response $r_{\xi}^{(2)}(t)$ and third-order response $r_{\xi}^{(3)}(t)$:

$$r_{\xi}(t) = r_{\xi}^{(2)}(t) + r_{\xi}^{(3)}(t).$$

The second- and third-order responses are defined as follows (see also Appendix 1):

$$r_\xi^{(2)}(t) = 2 \sum_{\tau_1, \tau_2} K_{LR}^{(2)}(\tau_1, \tau_2) s_\xi(t - \tau_1) s_{\xi+1}(t - \tau_2)(\Delta t)^2,$$

$$r_\xi^{(3)}(t) = 3 \sum_{\tau_1, \tau_2, \tau_3} K_{LRL}^{(3)}(\tau_1, \tau_2, \tau_3) \left[ \left( s_\xi(t - \tau_1) s_{\xi+1}(t - \tau_2) s_\xi(t - \tau_3) - s_{\xi+1}(t - \tau_1) s_\xi(t - \tau_2) s_{\xi+1}(t - \tau_3) \right) \right] (\Delta t)^3,$$

where $s_\xi(t)$ and $s_{\xi+1}(t)$ denote the visual inputs that the EMD at spatial location $\xi$ receives. (In *Figure 2B* and *Figure 2—figure supplement 1*, we used $s_L(t)$ and $s_R(t)$ to represent $s_\xi(t)$ and $s_{\xi+1}(t)$). The second-order response $r_\xi^{(2)}(t)$ is the sum of second-order features, $s_\xi(t - \tau_1) s_{\xi+1}(t - \tau_2)$, weighted by the second-order kernel, $K_{LR}^{(2)}(\tau_1, \tau_2)$. The third-order response, $r_\xi^{(3)}(t)$, is the sum of third-order features, $s_\xi(t - \tau_1) s_{\xi+1}(t - \tau_2) s_\xi(t - \tau_3)$, and $s_{\xi+1}(t - \tau_1) s_\xi(t - \tau_2) s_{\xi+1}(t - \tau_3)$, weighted by the third-order kernel, $K_{LRL}^{(3)}(\tau_1, \tau_2, \tau_3)$. Note that we use the notation $K_{LLR}^{(3)}$ in Appendix 1, which can be transformed into $K_{LRL}^{(3)}$ by interchanging the position of the second and the third spatial and temporal arguments. In particular, $K_{LRL}^{(3)}(\tau_1, \tau_2, \tau_3) = K_{LLR}^{(3)}(\tau_1, \tau_3, \tau_2)$.

## Measuring Volterra kernels with stochastic stimuli and reverse-correlation

To estimate the kernels, we presented stochastic binary stimuli to flies and reverse-correlated the corresponding response with the input. In particular, we estimated the second-order kernel by reverse-correlating the mean-subtracted turning response with the products of two points in space and time (Appendix 1),

$$\hat{K}_{LR}^{(2)}(\tau_1, \tau_2) = \frac{1}{2} \left( \hat{K}_{LR-3}^{(2)}(\tau_1, \tau_2) + \hat{K}_{LR-4}^{(2)}(\tau_1, \tau_2) \right),$$

where $\hat{K}_{LR-3}^{(2)}(\tau_1, \tau_2)$ is the estimated second-order kernel from the three-bar-block stimulus and $\hat{K}_{LR-4}^{(2)}(\tau_1, \tau_2)$ is the estimated second-order kernel from the four-bar-block stimulus. In particular,

$$\hat{K}_{LR-3(4)}^{(2)}(\tau_1, \tau_2) = \frac{1}{54} \frac{1}{2\gamma_{\text{stim}}^4(\Delta t)^2} \sum_{\xi=1,2,3,(4)} \frac{1}{T} \sum_t r_{\text{turn}}(t) s_\xi(t - \tau_1) s_{\xi+1}(t - \tau_2),$$

where $r_{\text{turn}}(t)$ is the mean-subtracted response, and $\gamma_{\text{stim}}$ is the magnitude of the contrast in the binary stimulus (Appendix 1). $s_1(t), s_2(t), s_3(t)$ represent the contrasts of 3 independent bars in 3-bar-block stimulus, and $s_{\xi+1}(t - \tau_2) \equiv s_1(t - \tau_2)$ for $\xi = 3$. Similarly, $s_1(t), s_2(t), s_3(t), s_4(t)$ represent the contrasts of 4 independent bars in four-bar-block stimulus, and $s_{\xi+1}(t - \tau_2) \equiv s_1(t - \tau_2)$ for $\xi = 4$. We presented three-bar-block stimulus to 35 flies, and four-bar-block stimulus to 37 flies, for $T = 20$ min. All fly kernel estimates were averaged to generate the final kernel estimate. Note that we enforce $\hat{K}_{LR}^{(2)}(\tau_1, \tau_2) = 0$ when $\tau_1 = \tau_2$, because we model the visual motion estimator as a mirror-antisymmetric operator (Appendix 1).

Similarly, we estimated the third-order kernel by reverse-correlating the mean-subtracted turning response with the products of three points in space and time (Appendix 1).

$$\hat{K}_{LRL}^{(3)}(\tau_1, \tau_2, \tau_3) = \frac{1}{2} \left( \hat{K}_{LRL-3}^{(3)}(\tau_1, \tau_2, \tau_3) + \hat{K}_{LRL-4}^{(3)}(\tau_1, \tau_2, \tau_3) \right),$$

$$\hat{K}_{RLR}^{(3)}(\tau_1, \tau_2, \tau_3) = \frac{1}{2} \left( \hat{K}_{RLR-3}^{(3)}(\tau_1, \tau_2, \tau_3) + \hat{K}_{RLR-4}^{(3)}(\tau_1, \tau_2, \tau_3) \right),$$

where

$$\hat{K}_{LRL-3(4)}^{(3)}(\tau_1, \tau_2, \tau_3) = \frac{1}{54} \frac{1}{6} \frac{1}{\gamma_{\text{stim}}^6(\Delta t)^3} \sum_{\xi=1,2,3,(4)} \frac{1}{T} \sum_t r_{\text{turn}}(t) s_\xi(t - \tau_1) s_{\xi+1}(t - \tau_2) s_\xi(t - \tau_3),$$

$$\hat{K}_{RLR-3(4)}^{(3)}(\tau_1,\tau_2,\tau_3) = \frac{1}{54}\frac{1}{6}\frac{1}{\gamma_{stim}^6(\Delta t)^3}\sum_{\xi=1,2,3,(4)}\frac{1}{T}\sum_t r_{turn}(t)s_{\xi+1}(t-\tau_1)s_\xi(t-\tau_2)s_{\xi+1}(t-\tau_3),$$

and $\tau_1 \neq \tau_3$.

We then enforced mirror anti-symmetry by,

$$\hat{K}_{LR-sym}^{(2)}(\tau_1,\tau_2) = \frac{1}{2}\left(\hat{K}_{LR}^{(2)}(\tau_1,\tau_2) - \hat{K}_{LR}^{(2)}(\tau_2,\tau_1)\right),$$

$$\hat{K}_{LRL-sym}^{(3)}(\tau_1,\tau_2,\tau_3) = \frac{1}{2}\left(\hat{K}_{LRL}^{(3)}(\tau_1,\tau_2,\tau_3) - \hat{K}_{RLR}^{(3)}(\tau_1,\tau_2,\tau_3)\right),$$

$$\hat{K}_{RLR-sym}^{(3)}(\tau_1,\tau_2,\tau_3) = -\hat{K}_{LRL-sym}^{(3)}(\tau_1,\tau_2,\tau_3),$$

where $\hat{K}_{LR-sym}^{(2)}(\tau_1,\tau_2)$, $\hat{K}_{LRL-sym}^{(3)}(\tau_1,\tau_2,\tau_3)$ and $\hat{K}_{RLR-sym}^{(3)}(\tau_1,\tau_2,\tau_3)$ are the symmetrized kernels, and we refer to them as the mirror anti-symmetric component.

We next evaluated how much variance in the fly's turning behavior can be explained by the estimated second- and third-order kernels. We presented the same stochastic stimulus sequence to many flies, so we averaged the turning response from different flies, denoted as $\bar{r}_{turn}(t)$, to estimate the true stimulus-associated turning response $r_{stim-driven}(t)$. We predicted the turning response $r_{pred}(t)$ to the same stimulus sequence,

$$r_{pred}(t) = r_{pred}^{(2)}(t) + r_{pred}^{(3)}(t),$$

where $r_{pred}^{(2)}(t)$ $(r_{pred}^{(3)}(t))$ is the predicted response from the second-order (third-order) kernel. We calculated $r_{pred}(t)$ using only the anti-symmetric component of the kernel, that is $\hat{K}_{LR-sym}^{(2)}(\tau_1,\tau_2)$ and $\hat{K}_{LRL-sym}^{(3)}(\tau_1,\tau_2,\tau_3)$. The Pearson correlation between $r_{pred}(t)$ and $\bar{r}_{turn}(t)$ were 0.686 and 0.787 in three-bar and four-bar experiments.

If fly turning responses are driven only by visual stimuli, are mirror anti-symmetric, and use only second- and third-order correlations, then the measured second- and third-order kernel would explain all the variance in the stimulus-driven turning responses, and the Pearson correlation between $r_{pred}(t)$ and $r_{stim-driven}(t)$ should be one. However, our measured kernels only explained about half of the variance. There are several potential reasons for this. First, the turning responses of flies appeared very noisy, making it difficult to estimate the true stimulus-driven response. That is, $\bar{r}_{turn}(t)$ was a poor estimation of $r_{stim-driven}(t)$. Second, we wanted to minimally extend the canonical second-order motion detector while being able to account for light-dark asymmetric visual processing, so we added only one more term, the third-order kernel. However, the fly might respond to higher order spatiotemporal correlations in visual inputs, and our model did not capture them.

## Representing the second- and third-order kernels in the impulse response format

To better understand and visualize the extracted kernels, we rearranged the elements in the kernels such that we could interpret kernels as the *impulse response* to a pair (triplet) of contrasts. This is analogous to the *impulse response* to a single contrast change at one point in a linear system.

Before rearrangement, the rows (columns) of the second-order kernel represent the temporal argument $\tau_1$ $(\tau_2)$ in the matrix $\hat{K}_{LR-sym}^{(2)}(\tau_1,\tau_2)$. After rearrangement, the rows correspond to the time since the more recent point, and the columns represent different temporal intervals between the two points, with negative intervals meaning that the right point is more recent than the left point (*Figure 4—figure supplement 2D*) (*Salazar-Gatzimas et al., 2016*). We denote this new format as

$$\hat{K}_{impulse}^{(2)}(\tau,\Delta\tau_{21}) \equiv \hat{K}_{LR-sym}^{(2)}(\tau,\tau+\Delta\tau_{21}),$$

where $\Delta\tau_{21} = \tau_2 - \tau_1$ and $\tau = \min(\tau_1,\tau_1+\Delta\tau_{21})$. Because we have enforced mirror anti-symmetry in $\hat{K}_{LR-sym}^{(2)}$, the columns of $\hat{K}_{impulse}^{(2)}(\tau,\Delta\tau_{21})$ are anti-symmetric around $\Delta\tau_{21}=0$. We interpreted the

columns of $\hat{K}^{(2)}_{impulse}(\tau, \Delta\tau_{21})$ as the impulse response of the fly to a pair of adjacent contrast changes separated by $\Delta\tau_{21}$ in time.

Similarly, before rearrangement, the three dimensions of the third-order kernel represent the three temporal arguments of $\hat{K}^{(3)}_{LRL-sym}(\tau_1, \tau_2, \tau_3)$. Once rearranged, we define

$$\hat{K}^{(3)}_{impulse}(\tau, \Delta\tau_{21}, \Delta\tau_{31}) \equiv \hat{K}^{(3)}_{LRL-sym}(\tau, \tau + \Delta\tau_{21}, \tau + \Delta\tau_{31}),$$

where $\tau = \min(\tau_1, \tau_1 + \Delta\tau_{21}, \tau_1 + \Delta\tau_{31})$, $\Delta\tau_{21} = \tau_2 - \tau_1$, and $\Delta\tau_{31} = \tau_3 - \tau_1$. Rows again represent the time since the last point, the columns represent the temporal distance between the more recent point on the left and the sole right point, and the third tensor dimension represents the temporal distance between two left points (*Figure 4B*). For this third-order kernel, we also summed along the rows for 0.75 s to define the summed kernel strength (*Figure 4C*),

$$\hat{K}^{(3)}_{summed}(\Delta\tau_{21}, \Delta\tau_{31}) = \sum_{\tau < 0.75} \hat{K}^{(3)}_{impulse}(\tau, \Delta\tau_{21}, \Delta\tau_{31}).$$

## Testing the significance of the measured third-order kernel with 'null kernels'

We tested the significance of the measured kernel with synthetic null kernels (*Figure 4C*). We shifted the stimulus with 100 random temporal offsets (the offset was at least 2 seconds long), reverse-correlated these shifted stimuli with responses, and generated 100 synthetic null kernels. The 100 kernels extracted from the misaligned stimulus and response were used to test the significance of the real kernel. We calculated the summed kernel strength of these 100 null kernels, and built the null distribution of summed kernel strength and performed two-tailed z-test. We tested kernel strength in the region of the kernels: $\tau_3 - \tau_1$ from 0 to 250 ms, and $\tau_2 - \tau_1$ from $-250$ to 250 ms, which equaled $528 = 16 \times 33$ kernel strengths in total. There are 43 significant (p <0.05) responses, and around 23% of the total significant points (10 in total) aggregated when $|\tau_1 - \tau_2| < 83$ ms, $|\tau_3 - \tau_1| < 83$ ms. Therefore, we further simplified our kernel by setting third-order kernel elements to zero when $|\tau_1 - \tau_2| \geq 83$ ms or $|\tau_3 - \tau_1| \geq 83$ ms, and denoted the 'cleaned' kernel as $\hat{K}^{(3)}_{LRL-sym-clean}(\tau_1, \tau_2, \tau_3)$. To be consistent, we also set elements of the second-order kernel to zero when $|\tau_1 - \tau_2| > 83$ ms, and denoted it as $\hat{K}^{(2)}_{LR-sym-clean}(\tau_1, \tau_2)$.

The exact p-values for the summed third-order strength were shown for $\tau_2 - \tau_1 = [-133\text{ms}, 133\text{ms}]$ with 16.7 ms increment in the following table from top to bottom.

| $\Delta\tau_{31} = 16$ ms | $\Delta\tau_{31} = 33.3$ ms | $\Delta\tau_{31} = 50$ ms |
|---|---|---|
| 0.4328 | 0.7506 | 0.3948 |
| 0.9784 | 0.0514 | 0.1870 |
| 0.7381 | 0.0587 | 0.0904 |
| 0.3340 | 0.9650 | 0.5661 |
| 0.4435 | 0.6801 | 0.6273 |
| 0.2154 | 0.8081 | 0.8081 |
| 0.0645 | 0.0942 | 0.0641 |
| 0.0008 | 0.7077 | 0.0015 |
| <0.0001 | 0.3014 | 0.9246 |
| <0.0001 | 0.0823 | 0.9375 |
| 0.8873 | <0.0001 | 0.0149 |
| 0.0177 | 0.3112 | 0.0001 |
| 0.1011 | 0.0487 | 0.2435 |
| 0.2565 | 0.3086 | 0.5983 |
| 0.6328 | 0.5057 | 0.7153 |
| 0.1037 | 0.4815 | 0.3233 |
| 0.3677 | 0.2075 | 0.6670 |

## Comparing the measured third-order kernel with the glider responses

We measured the fly's sensitivity to three-point correlations using a suite of third-order glider stimuli. Overall, we presented $52 = 13 \times 2 \times 2$ different stimuli (See Visual stimuli in Materials and methods, *Table 1*, *Figure 4—figure supplement 1A*). Each glider stimulus elicited sustained turning responses (*Clark et al., 2014*), so we averaged the response over time and denote it as $r^{\text{glider}}_{(\Delta\tau_{31},\Delta\tau_{21},L/R,P)}$, where the subscript specifies the stimulus type. Since we assume the fly's motion computation is mirror anti-symmetric, we subtracted responses to the pairs of gliders with different directions but with the same temporal interval and polarity, and denote it as $r^{\text{glider}}_{(\Delta\tau_{31},\Delta\tau_{21},P)}$,

$$r^{\text{glider}}_{(\Delta\tau_{31},\Delta\tau_{21},P)} = \frac{1}{2}\left(r^{\text{glider}}_{(\Delta\tau_{31},\Delta\tau_{21},O,P)} - r^{\text{glider}}_{(\Delta\tau_{31},\Delta\tau_{21},-O,P)}\right),$$

where $O$ denotes the orientation of the glider (i.e. left or right), and $-O$ denotes the opposite orientation. We plotted 18 out of 26 averaged responses in *Figure 4—figure supplement 1B*.

In *Table 1*, we listed the number of flies tested for each glider ($n_{P=1}$ is the number of flies tested with positive gliders, $n_{P=-1}$ with negative gliders), and the p-values of Student t-tests, which were tested against zero response ($p_{P=1}$ is the significance level for positive gliders, $p_{P=-1}$ for negative gliders).

The measured third-order kernel and the measured glider responses should both reflect the fly's sensitivity to three-point correlations. To test agreement between these two measurements, we used the measured third-order kernel to predict the fly's responses to glider stimuli. We made the prediction by summing the 'diagonal line' of the third-order kernel. Specifically, we found the predicted response to specific third-order gliders by summing over all elements in the kernel with the same temporal differences as the glider:

$$r^{(3)-\text{pred}}_{(\Delta\tau_{31},\Delta\tau_{21})} = 54 \times 6 \times \gamma^6_{stim} \sum_\tau K^{(3)}_{LRL-sym}(\tau, \tau + \Delta\tau_{21}, \tau + \Delta\tau_{31})(\Delta t)^3.$$

The constant of $54 \times 6$ takes into consideration the spatial summation all 54 putative EMDs and all six parts of third-order kernel in one EMD (Appendix 1), and $\gamma_{stim} = 1$ is the contrast of the glider stimuli. For gliders who have two points on the right side, we used $K_{RLR-sym}$ instead of $K_{LRL-sym}$. Since the third-order kernel is agnostic to the polarity of the three-point correlations and reflected only the average of the fly's sensitivity to positive and negative correlations, we averaged the responses of positive and negative gliders. This neglects higher-order components of glider responses that could nevertheless be biologically meaningful.

$$r^{\text{glider}-\text{ave}}_{(\Delta\tau_{31},\Delta\tau_{21})} = \frac{1}{2}\left(r^{\text{glider}}_{(\Delta\tau_{31},\Delta\tau_{21},1)} - r^{\text{glider}}_{(\Delta\tau_{31},\Delta\tau_{21},-1)}\right).$$

We then compared the $r^{\text{glider}-\text{ave}}_{(\Delta\tau_{31},\Delta\tau_{21})}$ with $r^{(3)-\text{pred}}_{(\Delta\tau_{31},\Delta\tau_{21})}$ (*Figure 4D*).

## Comparing the temporal structure of the second- and third-order kernels

To compare the temporal structure of the two kernels, we first rearranged and combined the elements in the third-order kernel into a two-dimensional representation and then rearrange the second-order kernel in the impulse response format. Specifically, we rearranged the elements in the third-order kernel into $\hat{K}^{(3)}_{aligned-LRL}(\tau, \Delta\tau_{LR}, \Delta\tau_{LL})$, where $\tau$ corresponds to the time since the most recent point, $\Delta\tau_{LR}$ is the average time difference between left points and the right point, and $2\Delta\tau_{LL}$ is the temporal separation between the two left points (*Figure 4—figure supplement 2A left*). In particular,

$$\hat{K}^{(3)}_{aligned-LRL}(\tau, \Delta\tau_{LR}, \Delta\tau_{LL}) \equiv \hat{K}^{(3)}_{LRL-sym}(\tau_2 + \Delta\tau_{LR} - \Delta\tau_{LL}, \tau_2, \tau_2 + \Delta\tau_{LR} + \Delta\tau_{LL}),$$

where $\tau = \min(\{\tau_2 + \Delta\tau_{LR} - \Delta\tau_{LL}, \tau_2, \tau_2 + \Delta\tau_{LR} + \Delta\tau_{LL}\})$. We similarly defined

$$\hat{K}^{(3)}_{aligned-RLR}(\tau, \Delta\tau_{RL}, \Delta\tau_{RR}) \equiv \hat{K}^{(3)}_{RLR-sym}(\tau_2 + \Delta\tau_{RL} - \Delta\tau_{RR}, \tau_2, \tau_2 + \Delta\tau_{RL} + \Delta\tau_{RR}),$$

where $\tau$ corresponds to the time since the most recent point, $\Delta\tau_{RL}$ is the average time difference between right points and the left point, and $2\Delta\tau_{RR}$ is the temporal separation between the two right points (*Figure 4—figure supplement 2A right*). Finally, we summed over the within-point time differences ($\Delta\tau_{RR}$ and $\Delta\tau_{LL}$), and summed these two pieces to obtain a matrix (*Figure 4—figure supplement 2B*),

$$\hat{K}^{(3)}_{align-2D}(\tau, \Delta\tau_{RL}) = \sum_{2\Delta\tau_{LL}<9\Delta t} \hat{K}^{(3)}_{aligned-LRL}(\tau, -\Delta\tau_{RL}, \Delta\tau_{LL}) + \sum_{2\Delta\tau_{RR}<9\Delta t} \hat{K}^{(3)}_{aligned-RLR}(\tau, \Delta\tau_{RL}, \Delta\tau_{RR}),$$

where $\Delta t = 1/60$ s.

$\hat{K}^{(3)}_{align-2D}(\tau, \Delta\tau_{RL})$ has rows and columns that are conceptually comparable to those of $K^{(2)}_{impulse}(\tau, \Delta\tau_{21})$, as rows represent times since the most recent point and columns describe the temporal distance between right and left points. However, in $\hat{K}^{(2)}_{impulse}$ the columns are spaced by 16.67 ms *Figure 4—figure supplement 2D*), whereas in $\hat{K}^{(3)}_{align-2D}$ the columns are spaced by 8.33 ms (*Figure 4—figure supplement 2B*). This results from the fact that $\Delta\tau_{21}$ is an integer number of frames in $\hat{K}^{(2)}_{impulse}$, whereas $2\Delta\tau_{RL}$ is an integer number of frames in $\hat{K}^{(3)}_{align-2D}$. We thus averaged two neighboring elements in $\hat{K}^{(3)}_{align-2D}$ (*Figure 4—figure supplement 2C*), so that it has the same resolution as the $\hat{K}^{(2)}_{impulse}$.

We then summed both $\hat{K}^{(3)}_{align-2D}$ and $\hat{K}^{(2)}_{impulse}$ in each column, and we rescaled the two summed kernels so that the norm of each was 1 (*Figure 4E*). In order to ease the visual comparison of the temporal structure between the two kernels, we also flipped the sign of the summed $\hat{K}^{(3)}_{align-2D}$.

## Comparing the second- and third-order kernels with the singular value decomposition (SVD)

We factorized $\hat{K}^{(3)}_{align-2D}$ and $\hat{K}^{(2)}_{impulse}$ into the products of a set of basis vectors with SVD (*Figure 4—figure supplement 2EFG*),

$$\hat{K}^{2}_{impulse} = U^{(2)} \cdot \sum{}^{(2)} \cdot V^{(2)T},$$

$$\hat{K}^{(3)}_{align-2D} = U^{(3)} \cdot \sum{}^{(3)} \cdot V^{(3)T},$$

where $U^{(i)}, \Sigma^{(i)}, V^{(i)}$, are the left-singular vectors, singular values, and right-singular vectors of the associated $i$th-order kernel. We use $u^{(2)}_1, v^{(2)}_1$ ($u^{(3)}_1, v^{(3)}_1$) to denote the left and right singular vectors corresponding to the largest singular values. For visualization purposes, in *Figure 4—figure supplement 2G*, we flipped the sign of $v^{(3)}_1$ so that readers could visually compare the temporal structure of these two vectors.

## Extracting kernels of various motion detectors that were optimized to predict image velocity in natural scenes

We characterized four other motion detectors (*Figure 4—figure supplement 3*) with Volterra kernels (*Fitzgerald and Clark, 2015*). These motion detectors have various physiological plausible structures and were optimized to predict image velocities in natural scenes. We fed the same stochastic binary stimuli sequence to these motion detectors, collected the corresponding responses, and extracted the second and third-order kernels using reverse-correlation. In *Figure 4F*, we presented the summed kernel strength of the third-order kernels. Note that we only presented several examples, and the spatiotemporal arguments of these examples are represented graphically.

## Natural scene dataset

We used a natural scene dataset (*Meyer et al., 2014*), which contains 421 panoramic luminance-calibrated naturalistic two-dimensional pictures. Each picture has $927 \times 251$ pixels and subtends 360° horizontally and 97.4° vertically, so that the spatial resolution is ~0.30°/pixel. In our study, we used 1-dimensional images, which were single rows from the two-dimensional pictures. Therefore, there were $105671 = 421 \times 251$ images in the dataset. We refer to the two-dimensional scenes as pictures or photographs, and refer to the one-dimensional slices as images.

## Preprocessing photographs

To simulate the photoreceptors, we converted the luminance pictures into contrast pictures with a blurring step and contrast computation step.

First, to simulate the spatial resolution of the fly's ommatidia, we blurred the original photograph (*Figure 4—figure supplement 1B*), denoted by $I$, with a two-dimensional Gaussian filter $f_{\mathrm{blur}}(x, y)$.

$$I_{\mathrm{blur}}(x, y) = I \star f_{\mathrm{blur}} = \sum_{u,v} I(x+u, y+v) f_{\mathrm{blur}}(u, v),$$

$$f_{\mathrm{blur}}(u, v) = \frac{1}{2\pi\lambda_{\mathrm{blur}}^2} \exp\left(-\frac{u^2+v^2}{2\lambda_{\mathrm{blur}}^2}\right), \ u_- \le u \le u_+, \ v_- \le v \le v_+.$$

where $\star$ denotes cross-correlation. The filter extends to $\pm 3\lambda_{\mathrm{blur}}$, that is $u_+ = v_+ = |u_-| = |v_-| = 3\lambda_{\mathrm{blur}}$, where $\lambda_{\mathrm{blur}}$ is related to full-width-at-half-maximum (FWHM) by $\lambda_{\mathrm{blur}} = \frac{\mathrm{FWHM}_{\mathrm{blur}}}{2\sqrt{2\ln 2}}$, and we chose $\mathrm{FWHM}_{\mathrm{blur}} = 5.3°$ (*Stavenga, 2003*). The original pictures cover the full circular range horizontally, but only 97.4° vertically. When the range of the filter extended beyond the vertical boundary of the picture, we padded the picture by 'vertical reflection'. This reflection padding was also used when we calculated the local mean-luminance. In Figure 1B, where we demonstrated one example picture, we did not perform this blurring step in order to preserve high spatial acuity such that it is pleasing for human eyes.

Second, we converted the luminance signals in the blurred photograph to contrast signals (*Figure 1—figure supplement 1CD*) (*Fitzgerald and Clark, 2015*),

$$c(x, y) = \frac{I_{\mathrm{blur}}(x, y) - I_{\mathrm{mean}}(x, y)}{I_{\mathrm{mean}}(x, y)},$$

$$I_{\mathrm{mean}}(x, y) = I_{\mathrm{blur}} \star f_{\mathrm{local-mean}},$$

where $c(x, y)$ is the contrast at each location $(x, y)$. $I_{\mathrm{mean}}$ is the local mean luminance, which is the averaged luminance weighted by a two-dimensional Gaussian spatial filter $f_{\mathrm{local-mean}}$. The length scale of $f_{\mathrm{local-mean}}$ can be equivalently described by $\lambda_{\mathrm{local-mean}}$ and $\mathrm{FWHM}_{\mathrm{local-mean}}$, where $\lambda_{\mathrm{local-mean}} = \frac{\mathrm{FWHM}_{\mathrm{local-mean}}}{2\sqrt{2\ln 2}}$. We swept $\mathrm{FWHM}_{\mathrm{local-mean}}$ from 10° to 75° (*Figure 3—figure supplement 2ABC*).

Alternatively, we also computed the local mean luminance over time instead of over space in *Figure 3—figure supplement 2FG*,

$$I_{\mathrm{mean}}(x, y, t) = \sum_{u} I_{\mathrm{blur-time}}(x, y, t) f_{\mathrm{local-mean-time}}(t - u),$$

where $I_{\mathrm{blur-time}}(x, y, t)$ is the simulated naturalistic moving luminance signal (*Fitzgerald and Clark, 2015*), and the local mean luminance was the averaged luminance signals over time, with the temporal filter $f_{\mathrm{local-mean-time}}(t) = \exp(-t/\tau_{\mathrm{local-mean}})$, where $f_{\mathrm{local-mean-time}}(t)$ is normalized to have a sum of 1. We swept $\tau_{\mathrm{local-mean}}$ from 10 ms to 500 ms (*Figure 3—figure supplement 2FG*).

Depending on the parameters in local mean computation, we had 20 natural scene datasets, including 14 datasets whose local mean luminance was computed statically and 6 dynamically. Unless specified, we used the natural scene dataset with contrast images preprocessed statically with $\mathrm{FWHM}_{\mathrm{local-mean}} = 25°$.

## Eliminating the higher order structure of natural scene ensemble

We created a synthetic image dataset where we effectively preserved only the second-order structure of natural scenes ensemble and eliminated the higher-order structure. We viewed each 1D image as a random vector determined by $\mathrm{P}_{\mathrm{natural}}(\boldsymbol{X})$, where $\boldsymbol{X}$ is a $n$-dimensional vector ($n = 927$ for 927 pixels). The covariance matrix of $\mathrm{P}_{\mathrm{natural}}(\boldsymbol{X})$ determines the point variance and pairwise spatial correlations in natural scenes. We intended to construct a Gaussian distribution $\mathrm{P}_{\mathrm{synthetic}}(\boldsymbol{X})$ such that its covariance matrix is the same as $\mathrm{P}_{\mathrm{natural}}(\boldsymbol{X})$. In this way, the image ensemble sampled from $\mathrm{P}_{\mathrm{synthetic}}(\boldsymbol{X})$ will contain the same second-order statistics as the natural scene ensemble, but it would lack the higher order statistics present in the natural scene ensemble.

Because the pixels in images are horizontally translational invariant, the covariance matrix of $\mathrm{P}_{\mathrm{natural}}(\boldsymbol{X})$ is a circulant matrix and can be diagonalized by a discrete Fourier transform. Therefore, we constructed $\mathrm{P}_{\mathrm{synthetic}}(\boldsymbol{X})$ and generated the synthetic dataset in the frequency domain (*Bialek, 2012*). Operationally, we first performed a discrete Fourier transform (with `fft` function in Matlab v2018a, RRID:SCR_001622) on each one-dimensional image in the natural scene dataset and obtained its Fourier domain representation $\boldsymbol{y}^{(i)} = \left( y_{k_1}^{(i)}, y_{k_2}^{(i)}, \ldots, y_{k_n}^{(i)} \right)$, where $(i)$ denotes the $i$th images, and $k_n$ denotes the $k_n$th Fourier component. We then calculated the average power of frequency $k_n$, denoted as $\sigma_{k_n}^2$, where $\sigma_{k_n}^2 = \frac{1}{m} \sum_i^m \left( y_{k_n}^{(i)} \right) \left( y_{k_n}^{(i)} \right)^*$, and $*$ denotes complex conjugation. At each frequency $k_n$, we built two Gaussian distributions $G_\Re \sim \mathcal{N}\left( 0, \frac{\sigma_{k_n}^2}{2} \right)$ and $G_\Im \sim \mathcal{N}\left( 0, \frac{\sigma_{k_n}^2}{2} \right)$. To generate one synthetic image, we sampled two real numbers from these two distributions as the real and imaginary part of Fourier component of the synthetic image at frequency $k_n$. In the end, we performed an inverse Fourier transform to the sampled Fourier components to gain a synthetic image in the spatial domain. In total, we generated 1000 high-order-structure-eliminated synthetic images, and refer to this synthetic image dataset to as the dataset in which higher order structure was eliminated.

## Computing and manipulating statistics of individual one-dimensional images

The natural scene dataset was comprised of an ensemble of heterogeneous images, and the statistics of different images can be drastically different from each other. Thus, we considered each one-dimensional image to have its own contrast distribution, $P_{\mathrm{pixel}}^{(i)}(X)$, where $i$ indexes the image and the contrast of each pixel in an image as an independent sample of the random variable $X$. For each natural image, we computed its sample mean, sample variance, sample skewness, and sample kurtosis of pixels, and show the histogram of these statistics (*Figure 5—figure supplement 1* and *Figure 5B*).

We generated $14 = 2 + 10 + 2$ synthetic image datasets to mimic various statistical properties of natural scenes (*Table 2*). These 14 datasets differ in three parameters: (1) the contrast range; (2) whether the contrast skewness was constrained; and (3) the specific value of the imposed skewness when the skewness was constrained. We conceptually justify this image synthesis method in Appendix 2, and here we focus on methodological details. For every image in the natural scene dataset, we generated a corresponding image in each synthetic image dataset. This involved two steps.

In step one, we determined all relevant image statistics and generated the corresponding maximum entropy distribution (MED). Operationally, for each individual image, we found its contrast range, $[c_{\min}, c_{\max}]$, the largest contrast magnitude, $\delta c = \max(|c_{\min}|, |c_{\max}|)$, and the sample mean, $c_\mu$. We then derived the contrast range specified in *Table 2*, binned the range into $N$ discrete levels, calculated the contrast frequency at each level to estimate $P_{\mathrm{pixel}}^{(i)}(X)$, and estimated the contrast variance $c_{\sigma^2}$ and skewness $c_\gamma$ from this estimated distribution. Finally, we solved the MED (Appendix 2) with the constrained statistics specified in *Table 2*. In the 'imposed skewness' column, 'NA' means that the skewness was not constrained in the MED.

In step two, we generated the synthetic image. The solved MED captures the pixel statistics but does not capture any spatial correlations between pixels. Therefore, we decided to interpolate between sampled pixels to coarsely mimic the spatial correlations. Operationally, for an individual image, we calculated its spatial correlation function and found the cutoff distance, $\Delta x_\alpha$, where the correlation falls below $\alpha = 0.2$. We then sampled contrast values independently from the solved

**Table 2.** Parameters for synthetic image datasets.

| Index of dataset | Imposed mean | Imposed variance | Imposed skewness | Contrast range | Discrete levels |
|---|---|---|---|---|---|
| MED-1 | $c_\mu$ | $c_{\sigma^2}$ | NA | $[c_\mu - \delta c, c_\mu + \delta c]$ | 32 |
| MED-2 | $c_\mu$ | $c_{\sigma^2}$ | $c_\gamma$ | $[c_\mu - \delta c, c_\mu + \delta c]$ | 32 |
| MED-3 | 0 | $c_{\sigma^2}$ | 1.25 | $[-\delta c, \delta c]$ | 32 |
| MED-4 | 0 | $c_{\sigma^2}$ | 1 | $[-\delta c, \delta c]$ | 32 |
| MED-5 | 0 | $c_{\sigma^2}$ | 0.75 | $[-\delta c, \delta c]$ | 32 |
| MED-6 | 0 | $c_{\sigma^2}$ | 0.5 | $[-\delta c, \delta c]$ | 32 |
| MED-7 | 0 | $c_{\sigma^2}$ | 0.25 | $[-\delta c, \delta c]$ | 32 |
| MED-8 | 0 | $c_{\sigma^2}$ | $-0.25$ | $[-\delta c, \delta c]$ | 32 |
| MED-9 | 0 | $c_{\sigma^2}$ | $-0.5$ | $[-\delta c, \delta c]$ | 32 |
| MED-10 | 0 | $c_{\sigma^2}$ | $-0.75$ | $[-\delta c, \delta c]$ | 32 |
| MED-11 | 0 | $c_{\sigma^2}$ | -1 | $[-\delta c, \delta c]$ | 32 |
| MED-12 | 0 | $c_{\sigma^2}$ | $-1.25$ | $[-\delta c, \delta c]$ | 32 |
| MED-13 | 0 | $c_{\sigma^2}$ | NA | [-2.5, 2.5] | 512 |
| MED-14 | 0 | $c_{\sigma^2}$ | $c_\gamma$ | [-2.5, 2.5] | 512 |

MED and placed these contrasts $\Delta x_\alpha$ pixels apart. Finally, we up-sampled the low-resolution image to a high-resolution image using the `resample` function in Matlab v2018a (RRID:SCR_001622).

Note that MED-3 through MED-7 are theoretically related to MED-8 through MED-12 by contrast inversion. We thus generated synthetic datasets MED-8 to MED-12 by simply inverting the contrast of images in dataset MED-3 to MED-7.

## Simulating naturalistic motion with natural scenes

To simulate the full-field motion signals induced by self-rotation in the natural environment, we rigidly translated images at various horizontal velocities (*Fitzgerald and Clark, 2015*). For each trial, we randomly chose one 1-dimensional image from the dataset. We had 35 image datasets in total, including 20 natural scene datasets preprocessed with different parameters and 15 synthetic image datasets, we therefore built 35 naturalistic motion datasets, where all images were sampled from a particular image dataset in each motion dataset. We sampled the velocity in two ways. First, we sampled it from a Gaussian distribution with zero mean and standard deviation of 114 °/s. This standard deviation is the measured standard deviation of the spontaneous rotational turning speed of freely walking flies. In *Figure 3—figure supplement 1*, we varied the standard deviation of the Gaussian distribution from 32 °/s to 512 °/s. Second, we selected an image velocity at discrete values ranging from 0 °/s to 1000 °/s with a 10 °/s interval. Given an image-velocity pair, we rigidly move this image with this velocity for one second. The temporal resolution is 60 Hz.

To eliminate any potential left-right asymmetry in naturalistic motion datasets, if we moved an image rightward at a certain speed, we always simulated a paired trial in which the same image was flipped horizontally and moved leftward with the same speed.

## Calculating the responses of different motion detectors to naturalistic motion signals

Our study concerns two motion detectors: the HRC and the measured kernels. For the HRC, we created an array of 54 overlapping HRC elementary motion detectors which extended 270° horizontally. We calculated the instantaneous HRCs' outputs at the end of each trial and averaged them across space to get the model's output $r_{\text{HRC}}$. For the measured kernels, we calculated the instantaneous output of the kernels at the end of each trial, including the second-order response, $r^{(2)}$, the third-order response, $r^{(3)}$, and the total response, $r = r^{(2)} + r^{(3)}$.

In *Table 3*, we listed the simulation parameters for each figure, including the motion detector, the image dataset, the velocity distribution, and the number of trials (image-velocity pairs). If the

**Table 3.** Natural scene datasets for naturalistic motion simulations.

| | Image dataset ($\mathrm{FWHM}_{\mathrm{local-mean}}/\tau_{\mathrm{local-mean}}$) | Velocity distribution ($\sigma_{\mathrm{vel}}$, or discrete values, °/s) | Number of trials | Motion Detector |
|---|---|---|---|---|
| *Figure 1C green* | Natural scene (25°) | Discrete [0:10:1000] | 1000 each velocity | HRC |
| *Figure 1C purple* | Synthetic-higher order structure eliminated | Discrete [0:10:1000] | 1000 each velocity | HRC |
| *Figure 3* | Natural scene (25°) | Gaussian (114) | 10000 | Fly |
| *Figure 3— figure supplement 1* | Natural scene (25°) | Gaussian (32, 64, 128, 256, 512) | 8000 each velocity distribution | Fly |
| *Figure 3— figure supplement 2DE* | Natural scene (10° ~ 75°) | Gaussian (114) | 8000 each $\mathrm{FWHM}_{\mathrm{local-mean}}$ | Fly |
| *Figure 3— figure supplement 2FG* | Natural scene (10 ~ 500 ms) | Gaussian (114) | 8000 each $\tau_{\mathrm{local-mean}}$ | Fly |
| *Figure 5CDE green, Figure 5— figure supplement 3A* | Synthetic-MED-1 | Gaussian (114) | 8000 | Fly |
| *Figure 5CDE brown Figure 5— figure supplement 3B* | Synthetic-MED-2 | Gaussian (114) | 8000 | Fly |
| *Figure 5FGH Figure 5— figure supplement 3CD* | Synthetic-MED 5–14 | Gaussian (114) | 8000 each dataset | Fly |
| *Figure 5—figure supplement 3G blue* | Synthetic-MED-13 | Gaussian (114) | 8000 | Fly |
| *Figure 5—figure supplement 3G red* | Synthetic-MED-14 | Gaussian (114) | 8000 | Fly |

dataset is natural scenes, we also listed its preprocessing parameter $\mathrm{FWHM}_{\mathrm{local-mean}}$. If the velocity distribution is Gaussian, we listed its standard deviation.

For example, in *Figure 3*, we predicted the motion estimates of the measured kernel to the naturalistic motion. The naturalistic motion was created with images sampled from the natural scene dataset that has preprocessing parameter $\mathrm{FWHM}_{\mathrm{local-mean}} = 25°$, and created with velocity sampled from the Gaussian distribution that has standard deviation of 114 °/s. We simulated 10,000 independent trials, and had 20,000 trials after enforcing left-right symmetry.

## Evaluating the performance of a motion detector

We denote the true velocity of the image as $v_{\mathrm{img}}$ and the response of a motion detector as $v_{\mathrm{est}}$. We assessed the accuracy of the motion estimation using two metrics. First, we measured the Pearson correlation $\rho$ between $v_{\mathrm{img}}$ and $v_{\mathrm{est}}$,

$$\rho = \frac{cov(v_{est}, v_{img})}{\sqrt{\mathrm{var}(v_{est})}\sqrt{\mathrm{var}(v_{img})}},$$

where the variances and covariances are evaluated across any of the naturalistic motion datasets introduced above. To estimate the uncertainty in $\rho$ induced by finite sample sizes, we randomly separated all independently simulated trials into 10 groups, calculated the Pearson correlation for each group, and estimated the *SEM* of the Pearson correlation across the groups. Second, when the image velocity was sampled at discrete values, we measured the variance of $v_{\mathrm{est}}$ conditional on each possible image velocity, $\mathrm{var}(v_{\mathrm{est}}|v_{\mathrm{img}} = v_0)$.

## Evaluating the improvement added by the third-order response

To evaluate how much improvement was added by the third-order response to the second-order response, we calculated the relative Pearson correlations: $\mathrm{improvement} = \frac{\rho^{(2+3)} - \rho^{(2)}}{\rho^{(2)}}$. As in *Evaluating the performance of a motion detector*, to estimate the uncertainty induced by finite sample sizes,

we separated all trials into 10 groups, calculated the Pearson correlation for each group, calculated the improvements in each group, and estimated the *SEM* of the improvements across the groups.

## Assessing the empirical weighting of the second-order and third-order responses

We modeled the image velocity as a linear combination of the second-order and third-order responses

$$v_{\text{img}} = \alpha^{(2)} r^{(2)} + \alpha^{(3)} r^{(3)} + \epsilon_{\text{img}},$$

and estimated the optimal weighting coefficients, $\left(\hat{\alpha}^{(2)}, \hat{\alpha}^{(3)}\right)$, using ordinary least square regression. We calculated $\rho_{\text{best}} = \frac{\text{cov}\left(r_{\text{best}}, v_{\text{img}}\right)}{\sqrt{\text{var}(r_{\text{best}})}\sqrt{\text{var}\left(v_{\text{img}}\right)}}$, where $r_{\text{best}} = \hat{\alpha}^{(2)} r^{(2)} + \hat{\alpha}^{(3)} r^{(3)}$. We computed the relative weighting coefficient as $w = \alpha^{(2)} / \alpha^{(3)}$.

## Calculating the residual of second-order response

The second-order response can be viewed as a function of the image velocity, as well as noise that depends on image structure:

$$r^{(2)} = \beta^{(2)} \, v_{\text{img}} + \epsilon^{(2)}.$$

We estimated the noise term as

$$\hat{\epsilon}^{(2)} = r^{(2)} - \hat{\beta}^{(2)} v_{\text{img}},$$

where $\hat{\beta}^{(2)}$ minimized the squared residual and $\hat{\epsilon}^{(2)}$ denotes the estimated residual (noise).

## Acknowledgements

The authors thanks Ann Hermundstad and Emily Cooper for comments on the manuscript. JEF was supported by the Swartz Foundation and the Howard Hughes Medical Institute. DAC and this research were supported by NIH R01EY026555, NIH P30EY026878, NSF IOS1558103, a Searle Scholar Award, a Sloan Fellowship in Neuroscience, the Smith Family Foundation, and the E Matilda Ziegler Foundation.

## Additional information

### Funding

| Funder | Grant reference number | Author |
| --- | --- | --- |
| National Institutes of Health | R01EY026555 | Juyue Chen<br>Damon A Clark |
| National Science Foundation | IOS1558103 | Juyue Chen<br>Damon A Clark |
| Chicago Community Trust | Searle Scholar Award | Holly B Mandel<br>Damon A Clark |
| Howard Hughes Medical Institute | | James E Fitzgerald |
| Alfred P. Sloan Foundation | Research Fellowship | Damon A Clark |
| The Swartz Foundation | | James E Fitzgerald |

The funders had no role in study design, data collection and interpretation, or the decision to submit the work for publication.

## Author contributions
Juyue Chen, Conceptualization, Software, Formal analysis, Validation, Investigation, Visualization, Methodology, Writing—original draft, Writing—review and editing; Holly B Mandel, Conceptualization, Software, Investigation, Methodology, Writing—review and editing; James E Fitzgerald, Conceptualization, Formal analysis, Supervision, Investigation, Visualization, Methodology, Writing—original draft, Writing—review and editing; Damon A Clark, Conceptualization, Supervision, Funding acquisition, Visualization, Methodology, Writing—original draft, Project administration, Writing—review and editing

## Author ORCIDs
Juyue Chen (iD) https://orcid.org/0000-0003-2176-4723
James E Fitzgerald (iD) https://orcid.org/0000-0002-0949-4188
Damon A Clark (iD) https://orcid.org/0000-0001-8487-700X

## Decision letter and Author response
Decision letter https://doi.org/10.7554/eLife.47579.sa1
Author response https://doi.org/10.7554/eLife.47579.sa2

# Additional files

## Supplementary files
• Transparent reporting form

## Data availability
All data and code to reproduce figures here are available at: https://github.com/ClarkLabCode/ThirdOrderKernelCode (copy archived at https://github.com/elifesciences-publications/ThirdOrderKernelCode). Data is also available on Dryad under https://doi.org/10.5061/dryad.7jm87bt.

The following dataset was generated:

| Author(s) | Year | Dataset title | Dataset URL | Database and Identifier |
| --- | --- | --- | --- | --- |
| Chen J, Mandel HB, Fitzgerald JE, Clark DA | 2019 | Data from: Asymmetric ON-OFF processing of visual motion cancels variability induced by the structure of natural scenes | https://doi.org/10.5061/dryad.7jm87bt | Dryad Digital Repository, 10.5061/dryad.7jm87bt |

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

## Appendix 1

## Modeling the fly's optomotor turning response with Volterra kernels

The visual motion system of the fly can generally be considered as a nonlinear mapping from spatiotemporal visual stimuli, denoted $s(x,t)$, to behavioral turning responses, denoted $r(t)$. In general, any time-invariant nonlinear system can be written with a Volterra series,

$$r(t) = H^{(0)} + H^{(1)}[s(x,t)] + H^{(2)}[s(x,t)] + H^{(3)}[s(x,t)] + \dots,$$

where each $H^{(n)}$ is a convolutional operator,

$$H^{(n)}[s(x,t)] = \int d\eta_1 \int d\tau_1 \cdots \int d\eta_n \int d\tau_n K^{(n)}(\eta_1,\dots,\eta_n,\tau_1,\dots,\tau_n) s(\eta_1,t-\tau_1) \cdots s(\eta_n,t-\tau_n),$$

$\eta_i$ are spatial coordinates, $\tau_i$ are temporal coordinates, and $K^{(n)}$ is termed the $n$th-order Volterra kernel (*Marmarelis and McCann, 1973*; *Schetzen, 1980*). The system is fully specified by these kernels. In this section, we describe how we use this framework to model the fly's visual system. We make several simplifying assumptions about the fly's visual system.

Assumption 1. The system is third-order. This implies, $K^{(n)}(\eta_1,\dots,\eta_n,\tau_1,\dots,\tau_n) = 0$ for $n>3$. This choice aims to account for the system's potential ability to asymmetrically process light-dark signals while minimally extending the canonical second-order algorithm.

Assumption 2. The system is spatially-invariant. This implies that

$$K^{(n)}(\eta_1,\dots,\eta_n,\tau_1,\dots,\tau_n) = K^{(n)}(\eta_1+\delta\eta,\dots,\eta_n+\delta\eta,\tau_1,\dots,\tau_n).$$

Assumption 3. The system consists of a spatial array of elementary motion detectors (EMDs), and each EMD responds to inputs that are close in space. Since the spatial distance between neighboring ommatidia in fruit fly is around 5°, we discretize space into pixels with $\Delta x = 5°$ resolution, and we use integer indices to denote the relative spatial location. We further assume that the fly EMD does not respond to interactions between points that are spaced more than $\Delta x$ apart,

$$K^{(n)}(\eta_1,\dots,\eta_n,\tau_1,\dots,\tau_n) = 0, \; \left|\eta_i - \eta_j\right| > \Delta x.$$

This modeling simplification is frequently made in fly neuroscience (*Buchner, 1976*), but modern receptive field measurements in individual motion detectors in Drosophila suggest that they integrate over a wider field of view (*Leong et al., 2016*; *Salazar-Gatzimas et al., 2016*).

Assumption 4. The system is mirror anti-symmetric. This means that if $r(t)$ is the system's response to stimulus $s(x,t)$, then the system will respond to $s(-x,t)$ with response $-r(t)$. This is an important assumption for motion processing systems, and we will discuss its implications for Volterra kernels later in this section.

Given the first three assumptions, we model the fly's turning response $r(t)$ as

$$r(t) = \sum_{\xi=1}^{N_{EMD}} \left( r_\xi^{(0)}(t) + r_\xi^{(1)}(t) + r_\xi^{(2)}(t) + r_\xi^{(3)}(t) \right),$$

where

$$r_\xi^{(0)}(t) = r_0,$$

$$r_\xi^{(1)}(t) = \int d\tau_1 K^{(1)}(\tau_1) s\left(x_\xi, t-\tau_1\right) + \int d\tau_1 K^{(1)}(\tau_1) s\left(x_\xi + \Delta x, t-\tau_1\right),$$

$$r_\xi^{(2)}(t) = \sum_{\eta_1 \in \{0, \Delta x\}} \sum_{\eta_2 \in \{0, \Delta x\}} \iint d\tau_1 d\tau_2 K^{(2)}(\eta_1, \eta_2, \tau_1, \tau_2) s(x_\xi + \eta_1, t - \tau_1) s(x_\xi + \eta_2, t - \tau_2),$$

$$r_\xi^{(3)}(t) = \sum_{\eta_1 \in \{0, \Delta x\}} \sum_{\eta_2 \in \{0, \Delta x\}} \sum_{\eta_3 = \{0, \Delta x\}} \iiint d\tau_1 d\tau_2 d\tau_3 K^{(3)}(\eta_1, \eta_2, \eta_3, \tau_1, \tau_2, \tau_3)$$
$$\cdot s(x_\xi + \eta_1, t - \tau_1) s(x_\xi + \eta_2, t - \tau_2) s(x_\xi + \eta_3, t - \tau_3).$$

The response $r(t)$ is the sum of responses contributed by the $i$th-order kernel at different spatial locations $\xi$, denoted with $r_\xi^{(i)}(t)$. $K^{(1)}(\tau_1)$, $K^{(2)}(\eta_1, \eta_2, \tau_1, \tau_2)$, and $K^{(3)}(\eta_1, \eta_2, \eta_3, \tau_1, \tau_2, \tau_3)$ approximates how each EMD maps the inputs $s(x_\xi, t)$ and $s(x_\xi + \Delta x, t)$ into its output. $N_{EMD} = 54$ is the number of putative EMD units.

Next, we will simplify the notations for $K^{(2)}(\eta_1, \eta_2, \tau_1, \tau_2)$ and $K^{(3)}(\eta_1, \eta_2, \eta_3, \tau_1, \tau_2, \tau_3)$ , and discuss symmetries in these kernels.

The second-order kernel $K^{(2)}(\eta_1, \eta_2, \tau_1, \tau_2)$ is a four-dimensional tensor, with 2 dimensions in time and 2 dimensions in space. Since $|\eta_1 - \eta_2| \leq \Delta x$ , the values that can be taken are limited 2 discrete points, and all possible combinations of $(\eta_1, \eta_2)$ are $(L, L)$, $(L, R)$, $(R, L)$ and $(R, R)$, where $L(R)$ denotes left (right). We replace the spatial arguments with subscripts, and the 4-dimensional $K^{(2)}(\eta_1, \eta_2, \tau_1, \tau_2)$ can be rewritten with four 2-dimensional kernels,

$$K^{(2)}(\eta_1, \eta_2, \tau_1, \tau_2) = \begin{cases} K_{LL}^{(2)}(\tau_1, \tau_2) \\ K_{LR}^{(2)}(\tau_1, \tau_2) \\ K_{RL}^{(2)}(\tau_1, \tau_2) \\ K_{RR}^{(2)}(\tau_1, \tau_2) \end{cases}.$$

In $K_{\eta_1, \eta_2}^{(2)}(\tau_1, \tau_2)$, the first (second) temporal argument is related to the spatial location represented by the first (second) subscript. For example, $K_{LR}^{(2)}(\tau_1, \tau_2)$ means $K^{(2)}(\eta_1 = L, \eta_2 = R, \tau_1, \tau_2)$.

There are three types of symmetries in these $K_{\eta_1, \eta_2}^{(2)}(\tau_1, \tau_2)$. First, since the system is translationally-invariant, $K_{LL}^{(2)}(\tau_1, \tau_2) = K_{RR}^{(2)}(\tau_1, \tau_2)$. Second, $K_{LR}^{(2)}(\tau_1, \tau_2)$ describes how the system responds to $s_L(t - \tau_1) s_R(t - \tau_2)$, and $K_{RL}^{(2)}(\tau_2, \tau_1)$ describes how the system responds to $s_R(t - \tau_2) s_L(t - \tau_1)$.

Because multiplication operator is commutative, that is
$s_L(t - \tau_1) s_R(t - \tau_2) = s_R(t - \tau_2) s_L(t - \tau_1)$, one should have $K_{LR}^{(2)}(\tau_1, \tau_2) = K_{RL}^{(2)}(\tau_2, \tau_1)$. Therefore, the second-order response of an EMD can be simplified as,

$$r_\xi^{(2)}(t) = \iint d\tau_1 d\tau_2 \big[ 2K_{LR}^{(2)}(\tau_1, \tau_2) s_\xi(t - \tau_1) s_{\xi+1}(t - \tau_2)$$
$$+ K_{LL}^{(2)}(\tau_1, \tau_2) \big( s_\xi(t - \tau_1) s_\xi(t - \tau_2) + s_{\xi+1}(t - \tau_1) s_{\xi+1}(t - \tau_2) \big) \big],$$

where we replace the spatial arguments in stimulus with discrete subscripts $\xi$ as well.

Third, to derive the consequences of mirror anti-symmetry, note that the second-order kernel is locally sensitive to two discrete points in space, $s(x, t) = [s_L(t), \; s_R(t)]$. Mirror-anti-symmetry states that if the system's response to $s_L(t) = a(t)$ and $s_R(t) = b(t)$ is $r(t)$, then the system's response to the reflected stimulus, $s_L(t) = b(t)$ and $s_R(t) = a(t)$, is $-r(t)$, where $a(t)$ and $b(t)$ are arbitrary functions of time. The system's response when $s_L(t) = a(t)$ and $s_R(t) = b(t)$ is

$$r_{\xi,ab}^{(2)}(t) = \iint d\tau_1 d\tau_2 \left[ K_{LL}^{(2)}(\tau_1, \tau_2)(a(t - \tau_1)a(t - \tau_2) + b(t - \tau_1)b(t - \tau_2)) + 2K_{LR}^{(2)}(\tau_1, \tau_2)a(t - \tau_1)b(t - \tau_2) \right],$$

and the response when $s_L(t) = b(t)$ and $s_R(t) = a(t)$ is

$$r^{(2)}_{\xi,ba}(t) = \iint d\tau_1 d\tau_2 \left[ K^{(2)}_{LL}(\tau_1,\tau_2)(b(t-\tau_1)b(t-\tau_2) + a(t-\tau_1)a(t-\tau_2)) + 2K^{(2)}_{LR}(\tau_1,\tau_2)b(t-\tau_1)a(t-\tau_2) \right].$$

If $r^{(2)}_{\xi,ba}(t) = -r^{(2)}_{\xi,ab}(t)$ for any $a(t)$ and $b(t)$, then one must have

$$K^{(2)}_{LL}(\tau_1,\tau_2) = -K^{(2)}_{LL}(\tau_1,\tau_2) = 0,$$

and

$$K^{(2)}_{LR}(\tau_1,\tau_2) = -K^{(2)}_{LR}(\tau_2,\tau_1).$$

Thus, the general expression for the second-order response of the EMD further simplifies to

$$r^{(2)}_{\xi}(t) = 2\iint d\tau_1 d\tau_2 K^{(2)}_{LR}(\tau_1,\tau_2) s_\xi(t-\tau_1) s_{\xi+1}(t-\tau_2).$$

We analyzed the third-order kernel in a similar manner. It is a 6-dimensional tensor, with 3 dimensions in time and 3 dimensions in space. Its spatial arguments are also limited to two points, and all possible combinations of $(\eta_1, \eta_2, \eta_3)$ are $(L,L,L)$, $(R,R,R)$, $(L,L,R)$, $(L,R,L)$, $(R,L,L)$, $(R,R,L)$, $(R,L,R)$ and $(L,R,R)$. We replace the spatial arguments with subscripts, and represent the 6-dimensional third-order kernel with eight 3-dimensional kernels,

$$K^{(3)}(\eta_1,\eta_2,\eta_3,\tau_1,\tau_2,\tau_3) = \begin{cases} K^{(3)}_{LLL}(\tau_1,\tau_2,\tau_3) \\ K^{(3)}_{RRR}(\tau_1,\tau_2,\tau_3) \\ K^{(3)}_{LLR}(\tau_1,\tau_2,\tau_3) \\ K^{(3)}_{LRL}(\tau_1,\tau_2,\tau_3) \\ K^{(3)}_{RLL}(\tau_1,\tau_2,\tau_3) \\ K^{(3)}_{RRL}(\tau_1,\tau_2,\tau_3) \\ K^{(3)}_{RLR}(\tau_1,\tau_2,\tau_3) \\ K^{(3)}_{LRR}(\tau_1,\tau_2,\tau_3) \end{cases}.$$

As was the case of the second-order kernel, in $K^{(3)}_{\eta_1,\eta_2,\eta_3}(\tau_1,\tau_2,\tau_3)$ the first (second, third) temporal argument is related to the spatial location denoted with the first (second, third) subscript. For example, $K^{(3)}_{LLR}(\tau_1,\tau_2,\tau_3) = K^{(3)}(\eta_1 = L, \eta_2 = L, \eta_3 = R, \tau_1,\tau_2,\tau_3)$.

As before, these eight blocks are redundant. From spatial-invariance and the commutative property of multiplication, one finds

$$K^{(3)}_{LLL}(\tau_1,\tau_2,\tau_3) = K^{(3)}_{RRR}(\tau_1,\tau_2,\tau_3),$$

$$K^{(3)}_{LLR}(\tau_1,\tau_2,\tau_3) = K^{(3)}_{LRL}(\tau_1,\tau_3,\tau_2) = K^{(3)}_{RLL}(\tau_3,\tau_1,\tau_2),$$

$$K^{(3)}_{RRL}(\tau_1,\tau_2,\tau_3) = K^{(3)}_{RLR}(\tau_1,\tau_3,\tau_2) = K^{(3)}_{LRR}(\tau_3,\tau_1,\tau_2).$$

For each 3D tensor, at least two out of the three spatial arguments are the same, so there is symmetry within the tensor. For example,

$$K^{(3)}_{LLR}(\tau_1,\tau_2,\tau_3) = K^{(3)}_{LLR}(\tau_2,\tau_1,\tau_3),$$

and

$$K^{(3)}_{LRR}(\tau_1,\tau_2,\tau_3) = K^{(3)}_{LRR}(\tau_1,\tau_3,\tau_2).$$

Following this logic, the third-order response of an EMD can be written as

$$r_\xi^{(3)}(t) = \iiint d\tau_1 d\tau_2 d\tau_3 \left[ 3K_{LLR}^{(3)}(\tau_1, \tau_2, \tau_3) s_\xi(t - \tau_1) s_\xi(t - \tau_2) s_{\xi+1}(t - \tau_3) \right.$$

$$+ 3K_{RRL}^{(3)}(\tau_1, \tau_2, \tau_3) s_{\xi+1}(t - \tau_1) s_{\xi+1}(t - \tau_2) s_\xi(t - \tau_3) + K_{LLL}^{(3)}(\tau_1, \tau_2, \tau_3)$$

$$\left. \cdot \left( s_\xi(t - \tau_1) s_\xi(t - \tau_2) s_\xi(t - \tau_3) + s_{\xi+1}(t - \tau_1) s_{\xi+1}(t - \tau_2) s_{\xi+1}(t - \tau_3) \right) \right].$$

One can again apply the definition of mirror anti-symmetry to find

$$K_{LLL}^{(3)}(\tau_1, \tau_2, \tau_3) = 0,$$

$$K_{LLR}^{(3)}(\tau_1, \tau_2, \tau_3) = -K_{RRL}^{(3)}(\tau_1, \tau_2, \tau_3).$$

Thus, with mirror anti-symmetry, the third-order response of an EMD can be simplified as,

$$r_\xi^{(3)}(t) = 3 \iiint d\tau_1 d\tau_2 d\tau_3 K_{LLR}^{(3)}(\tau_1, \tau_2, \tau_3) [(s_\xi(t - \tau_1) s_\xi(t - \tau_2 t) s_{\xi+1}(t - \tau_3) - s_{\xi+1}(t - \tau_1) s_{\xi+1}(t - \tau_2) s_\xi(t - \tau_3))].$$

We have shown how mirror anti-symmetry manifest in the second and third-order kernels. In a similar way, for the zero- and first-order kernel, mirror anti-symmetry implies that,

$$r_0 = 0,$$

$$K^{(1)}(\tau_1) = 0.$$

Intuitively, $r_0$ term is the response of the EMD when there is no visual input. A reasonable motion detector should not be biased to turn left or right when there is no visual input. The first-order kernel describes how the response is influenced by stimulus at a single spatial location, which should not give any motion information, thus $K^{(1)}(\tau_1) = 0$.

In summary, in our model, the turning response is

$$r(t) = \sum_\xi r_\xi(t) = \sum_\xi \left( r_\xi^{(2)}(t) + r_\xi^{(3)}(t) \right),$$

$$r_\xi^{(2)}(t) = 2 \iint d\tau_1 d\tau_2 K_{LR}^{(2)}(\tau_1, \tau_2) s_\xi(t - \tau_1) s_{\xi+1}(t - \tau_2),$$

$$r_\xi^{(3)}(t) = 3 \iiint d\tau_1 d\tau_2 d\tau_3 K_{LLR}^{(3)}(\tau_1, \tau_2, \tau_3) [(s_\xi(t - \tau_1) s_\xi(t - \tau_2) s_{\xi+1}(t - \tau_3) - s_{\xi+1}(t - \tau_1) s_{\xi+1}(t - \tau_2) s_\xi(t - \tau_3))],$$

where $r_\xi(t)$ is the response of each EMD at spatial location $\xi$. In the experiments, we discretized time into bins of size $\Delta t = 1/60$ seconds, so the integrals become summations, and we write the local responses as

$$r_\xi^{(2)}(t) = 2 \sum_{\tau_1, \tau_2} K_{LR}^{(2)}(\tau_1, \tau_2) s_\xi(t - \tau_1) s_{\xi+1}(t - \tau_2)(\Delta t)^2,$$

$$r_\xi^{(3)}(t) = 3 \sum_{\tau_1, \tau_2, \tau_3} K_{LLR}^{(3)}(\tau_1, \tau_2, \tau_3) [(s_\xi(t - \tau_1) s_\xi(t - \tau_2) s_{\xi+1}(t - \tau_3) - s_{\xi+1}(t - \tau_1) s_{\xi+1}(t - \tau_2) s_\xi(t - \tau_3))](\Delta t)^3.$$

## Appendix 2

# Manipulating image statistics with maximum entropy distributions

To investigate how the skewness of the image contrast distribution influences motion detection, we want to develop a method for generating synthetic images with specified skewness values. In Section 1, we formalize this image synthesis problem and provide a rationale for the maximum entropy method. In Section 2, we derive formulae for finding maximum entropy distributions (MED) with constrained lower-order moments. In Section 3, we discuss how the contrast range sets implicit constraints on the MED.

### 1. Motivation

We consider each 1-dimensional natural image to have its own contrast distribution, $P_{pixel}^{(i)}(X)$, such that the contrast of each pixel in the $i^{th}$ image is an independent sample from $P_{pixel}^{(i)}(X)$. We would like to generate a synthetic contrast distribution, $P_{syn}^{(i)}(X)$, such that $P_{syn}^{(i)}(X)$ shares certain low-order contrast statistics with $P_{pixel}^{(i)}(X)$. More specifically, we want $P_{syn}^{(i)}(X)$ and $P_{pixel}^{(i)}(X)$ to have matched means, variances, and/or skewness levels. However, it's important to recognize that $P_{syn}^{(i)}(X)$ is ambiguously determined by its lower-order moments, because many distributions share finite set of moments. Nevertheless, among all of these qualified distributions, the distribution with maximal entropy is unique. The maximum entropy requirement thus implicitly specifies all unconstrained statistics. Maximum entropy distributions are also conceptually appealing because they have the least statistical structure consistent with the set of chosen constraints. Here we use maximum entropy distributions to generate synthetic images with specific statistics by sampling pixel contrasts from $P_{syn}^{(i)}(X)$. By manipulating the statistics that define $P_{syn}^{(i)}(X)$, we are able to manipulate the contrast statistics of the synthetic image.

### 2. Solving maximum entropy distribution by minimizing the free energy function

To illustrate the structure of the maximum entropy distribution (MED) with constrained moments, we concretely consider the example when the MED is constrained to have a specific mean, variance, and skewness. We refer to this distribution as the mean-variance-and-skewness constrained MED. Since constraining the mean, variance, and skewness is equivalent to constraining the first, second and third moments, we equivalently refer to this distribution as the third-order MED.

The entropy of a random variable $X$ is

$$H(X) = \mathrm{E}[-\log P(X)],$$

where $P(X)$ is the probability distribution of $X$, and $\mathrm{E}[]$ is the expectation operator over $P(X)$. In particular, $\mathrm{E}[] = \int dx P(X = x)[]$. By definition, the third-order MED is the probability distribution, $P^*(X)$, such that

$$P^*(X) = \underset{P(X)}{\mathrm{argmax}}\, H(X),$$

subject to,

$$\mathrm{E}\left[X^i\right] = \mu_i,\ i = 1, 2, 3.$$

where $\mu_1$, $\mu_2$ and $\mu_3$ are the first-, second-, and third moments. To optimize for $P^*(X)$, one can introduce Lagrange multipliers, $\lambda_0, \lambda_1, \lambda_2$ and $\lambda_3$, to enforce the normalization of the probability

distribution and the three moment constraints. The problem is thus transformed into extremizing

$$L(P(X),\lambda_0,\lambda_1,\,\lambda_2,\,\lambda_3) = \mathrm{E}[-\log P(X)] + \lambda_0\left(\int dx P(X=x) - 1\right) + \sum_{i=1,2,3}\lambda_i\big(\mathrm{E}\big[X^i\big] - \mu_i\big).$$

Note that $L(P(X),\lambda)$ is a functional of $P(X)$, and a necessary condition for $P(X)$ to extremize $L$ is

$$0 = \frac{\delta L}{\delta P(X=x)} = -(\log P(X=x) + 1) + \lambda_0 + \lambda_1 x + \lambda_2 x^2 + \lambda_3 x^3.$$

This implies that third-order MED has the form

$$P(X=x) = \exp\big(-1 + \lambda_0 + \lambda_1 x + \lambda_2 x^2 + \lambda_3 x^3\big),$$

and the Lagrange multipliers must be solved to satisfy $\int dx P(X=x) = 1$, and $\mathrm{E}[X^i] = \mu_i,\ i=1,2,3$.

An alternative to solving the nonlinear constraint-satisfaction equations for the Lagrange multipliers is to find $\lambda^* = \big(\lambda_1^*,\lambda_2^*,\lambda_3^*\big)$ such that

$$\lambda^* = \underset{\lambda}{\mathrm{argmin}}\ F(\lambda),$$

where

$$F(\lambda) = \log(\mathrm{Z}(\lambda)),\, \mathrm{Z}(\lambda) = \int dx Q(x,\,\lambda),\, Q(x,\,\lambda) = \exp\big(\lambda_1(x - \mu_1) + \lambda_2\big(x^2 - \mu_2\big) + \lambda_3\big(x^3 - \mu_3\big)\big).$$

Here $Q(x,\lambda)$ can be thought of an unnormalized probability distribution for $X$, $\mathrm{Z}(\lambda)$ is the normalizing factor, and $F(\lambda)$ is the log of this normalizing factor. Readers familiar with statistical mechanics will recognize $F(\lambda)$ as the free energy function and $\mathrm{Z}(\lambda)$ as the partition function. It turns out that the third-order MED is

$$P^*(X) = \frac{1}{\mathrm{Z}(\lambda^*)} Q(x,\lambda^*).$$

This distribution manifestly has the right functional form, so to prove that it is the third-order MED we merely need to show that all constraints are satisfied by the distribution. Indeed, the constraint that $\int dx P(X=x) = 1$ is trivially satisfied because the partition function $\mathrm{Z}(\lambda^*)$ normalizes $Q(x,\lambda^*)$. Since $\lambda^*$ minimizes the free energy, we also know

$$\frac{\partial F(\lambda^*)}{\partial \lambda_i} = 0.$$

This derivative is easily evaluated as

$$\frac{\partial F(\lambda)}{\partial \lambda_i} = \frac{\partial \log(\mathrm{Z}(\lambda))}{\partial \lambda_i} = \frac{1}{\mathrm{Z}(\lambda)}\frac{\partial \mathrm{Z}(\lambda)}{\partial \lambda_i} = \frac{1}{\mathrm{Z}(\lambda)}\frac{\partial \int dx Q(x,\,\lambda)}{\partial \lambda_i} = \frac{1}{\mathrm{Z}(\lambda)}\int dx \frac{\partial Q(x,\,\lambda)}{\partial \lambda_i}$$

$$= \frac{1}{\mathrm{Z}(\lambda)}\int dx Q(x,\,\lambda)\big(x^j - \mu_i\big) = E_\lambda\big[X^i\big] - \mu_i,$$

where $E_\lambda[\,]$ is the expectation operator over $P_\lambda(x) = Q(x,\lambda)/\mathrm{Z}(\lambda)$. Consequently,

$$\frac{\partial F(\lambda^*)}{\partial \lambda_i} = 0 \Longrightarrow E_{\lambda^*}\big[X^i\big] = \mu_i,$$

and minimizing the free energy provides parameters that guarantee that the constraints are satisfied.

This free energy formulation is computationally convenient because $F(\lambda)$ is a convex function that can be easily minimized using powerful techniques from convex optimization. To see this, note that the matrix of second derivatives is,

$$\frac{\partial^2 F(\lambda)}{\partial\lambda_i\partial\lambda_j} = \frac{\partial}{\partial\lambda_i}\frac{1}{Z(\lambda)}\frac{\partial Z(\lambda)}{\partial\lambda_j} = -\left(\frac{1}{Z(\lambda)}\right)^2\frac{\partial Z(\lambda)}{\partial\lambda_i}\frac{\partial Z(\lambda)}{\partial\lambda_j} + \frac{1}{Z(\lambda)}\frac{\partial^2 Z(\lambda)}{\partial\lambda_i\partial\lambda_j}.$$

We showed above that

$$\frac{1}{Z(\lambda)}\frac{\partial Z(\lambda)}{\partial\lambda_i} = E_\lambda\left[x^i - \mu_i\right] \Longrightarrow -\left(\frac{1}{Z(\lambda)}\right)^2\frac{\partial Z(\lambda)}{\partial\lambda_i}\frac{\partial Z(\lambda)}{\partial\lambda_j} = -E_\lambda\left[x^i - \mu_i\right]E_\lambda\left[x^j - \mu_j\right].$$

Similarly, we next note that

$$\frac{1}{Z(\lambda)}\frac{\partial^2 Z(\lambda)}{\partial\lambda_i\partial\lambda_j} = \frac{1}{Z(\lambda)}\frac{\partial^2}{\partial\lambda_i\partial\lambda_j}\int dx Q(x,\lambda) = \frac{1}{Z(\lambda)}\int dx\frac{\partial^2 Q(x,\lambda)}{\partial\lambda_i\partial\lambda_j}$$
$$= \frac{1}{Z(\lambda)}\int dx Q(x,\lambda)\left(x^i - \mu_i\right)\left(x^j - \mu_j\right) = E_\lambda\left[\left(x^i - \mu_i\right)\left(x^j - \mu_j\right)\right].$$

Therefore, these two terms together imply that

$$\frac{\partial^2 F(\lambda)}{\partial\lambda_i\partial\lambda_j} = E_\lambda\left[\left(x^i - \mu_i\right)\left(x^j - \mu_j\right)\right] - E_\lambda\left[x^i - \mu_i\right]\mathrm{E}\left[x^j - \mu_j\right] = \mathrm{Cov}_\lambda\left[x^i - \mu_i, x^j - \mu_j\right].$$

Since the second derivative of $F(\lambda)$ is a covariance matrix for all values of $\lambda$, and all covariance matrices are positive semi-definite, this implies $F(\lambda)$ is a convex function.

Similarly, one can find the maximum entropy distribution (MED) with constrained mean and variance by minimizing another free energy function:

$$\lambda^* = \left(\lambda_1^*, \lambda_2^*\right) = \underset{\lambda}{\mathrm{argmin}}\, F(\lambda_1, \lambda_2),$$

where

$$F(\lambda) = \log(Z(\lambda)), Z(\lambda) = \int dx Q(x,\lambda), Q(x,\lambda) = \exp(\lambda_1(x - \mu_1) + \lambda_2(x - \mu_2)),$$

and the maximum entropy distribution will be

$$P^*(X) = \frac{1}{Z(\lambda^*)}Q(x,\lambda^*).$$

We refer this distribution as the mean-and-variance-constrained MED or second-order MED.

In practice, we solved each MED by numerically minimizing its associated free energy function with the **fminunc** function in MATLAB v2018a.

## 3. Contrast bounds set implicit constraints on the MED

In the previous section, we emphasized that the free energy, $F$, depends on the Lagrange multipliers parameters $\lambda$, because optimizing over these parameters allowed us to find the corresponding MED. However, the free energy also depends on the constrained moments, $\mu$, which entered the formulae as a set of fixed parameters. Moreover, the free energy depends on the state space of the random variable, which is the set of values that $X$ can take. For continuous variables, this manifests as the integral bound, $[a, b]$, in the partition function, $Z(\lambda) = \int_a^b dx Q(x, \lambda)$. Therefore, we generally expect

$$\lambda^* = \lambda^*(\mu, a, b) = \underset{\lambda}{\mathrm{argmin}}\, F(\lambda | \mu, a, b).$$

Since $X$ represents the pixel contrast, we refer to $[a, b]$ as the contrast range and $a$ or $b$ separately as contrast bounds. In this section, we discuss in detail how $\lambda^*$ depends on $[a, b]$. Since $[a, b]$ influences the MED, we say $[a, b]$ sets an implicit constraint on the MED.

A natural choice of the contrast range is the entire real line. For example, when one constrains the mean and variance of a distribution and defines $X$ on the entire real line, then

the MED is the Gaussian distribution. Moreover, by rewriting the probability density of the Gaussian

$$P(X = x) = \frac{1}{\sqrt{2\pi\sigma^2}}\exp\left(-\frac{(x-\mu)^2}{2\sigma^2}\right),$$

in a slightly different form

$$P(X = x) = \frac{e^{-1/2}}{\sqrt{2\pi\sigma^2}}\exp\left(\frac{\mu}{\sigma^2}(x-\mu) - \frac{1}{2\sigma^2}(x^2 - \sigma^2 - \mu^2)\right),$$

we may identify the Lagrange multipliers that minimize the free energy as, $\lambda_1^* = \frac{\mu}{\sigma^2}$, $\lambda_2^* = -\frac{1}{2\sigma^2}$.

In the Gaussian distribution, the probability density drops to zero very fast as $x$ goes to positive or negative infinity, because $\lambda_2 x^2$ is a large negative number in either case. However, when one wants to find an MED whose highest-order constraint is an odd-order moment, a finite contrast bound becomes necessary. This is because the sign of $\lambda_{2n+1}x^{2n+1}$ depends on the sign of $x$ for any natural number $n$, thereby causing the probability density to approach zero on one half of the real line and to explode on the other half. To illustrate this point concretely, consider the first-order MED that has a constrained mean. The first-order MED must take the form

$$P(X = x) = \frac{1}{Z}\exp(\lambda_1(x-\mu_1)).$$

In this distribution, $\exp(\lambda_1(x-\mu_1))$ blows up when $x$ either goes to $+\infty$ or $-\infty$ (unless $\lambda_1 = 0$). In any case, no normalizing factor $Z$ could satisfy $\int_{-\infty}^{+\infty}\frac{1}{Z}\exp(\lambda_1(x-\mu_1)) = 1$. Therefore, the first-order MED is generally ill-defined on the real line.

We thus sought to compute the dependence of the first-order MED on its contrast range. The parameters in the MED minimize the free energy

$$\lambda^* = \underset{\lambda}{\arg\min}\, F(\lambda|a,b,\mu_1) = \underset{\lambda}{\arg\min}\, \log \int_a^b dx \exp(\lambda(x-\mu_1)).$$

Without loss of generality, we can shift the $x$-axis such that the contrast range is symmetric around 0. Setting $x' = x - \frac{1}{2}(b+a)$, we find

$$\lambda^* = \underset{\lambda}{\arg\min}\, F(\lambda|b',\mu') = \underset{\lambda}{\arg\min}\, \log \int_{-b'}^{b'} dx' \exp(\lambda(x'-\mu')),$$

where $b' = \frac{1}{2}(b-a)$ and $\mu' = \mu_1 - \frac{1}{2}(b+a)$.

This optimization problem for the first-order MED can be solved analytically. First, note that the partition function is

$$Z(\lambda) = \int_{-b'}^{b'} dx' \exp(\lambda(x'-\mu')) = \begin{cases} 2b' & \lambda = 0 \\ \frac{1}{\lambda}\left(e^{\lambda(b'-\mu')} - e^{\lambda(-b'-\mu')}\right) & \lambda \neq 0 \end{cases}.$$

This partition function is continuous at $\lambda = 0$. To see this, note that

$$\lim_{\lambda\to 0} Z(\lambda) = \lim_{\lambda\to 0} \frac{e^{\lambda(b'-\mu')} - e^{\lambda(-b'-\mu')}}{\lambda} = \lim_{\lambda\to 0} \frac{(b'-\mu')e^{\lambda(b'-\mu')} - (-b'-\mu')e^{\lambda(-b'-\mu')}}{1} = 2b' = Z(0),$$

where we used L'Hôpital's rule to evaluate the limit. Therefore the free energy,

$$F(\lambda|b',\mu') = \begin{cases} \log(2b') & \lambda = 0 \\ \log\left(e^{\lambda(b'-\mu')} - e^{\lambda(-b'-\mu')}\right) - \log(\lambda) & \lambda \neq 0 \end{cases},$$

is also continuous at $\lambda = 0$. Furthermore, the partition function is differentiable at $\lambda = 0$. In particular,

$$\frac{\mathrm{d}Z}{\mathrm{d}\lambda}\bigg|_{\lambda=0} = \lim_{h\to 0}\frac{Z(h)-Z(0)}{h}$$

$$= \lim_{h\to 0}\frac{\frac{1}{h}\left(e^{h(b'-\mu')}-e^{h(-b'-\mu')}\right)-2b'}{h}$$

$$= \lim_{h\to 0}\frac{e^{h(b'-\mu')}-e^{h(-b'-\mu')}-2b'h}{h^2}$$

$$= \lim_{h\to 0}\frac{(b'-\mu')e^{h(b'-\mu')}-(-b'-\mu')e^{h(-b'-\mu')}-2b'}{2h}$$

$$= \lim_{h\to 0}\frac{(b'-\mu')^2 e^{h(b'-\mu')}-(-b'-\mu')^2 e^{h(-b'-\mu')}}{2}$$

$$= -2b'\mu',$$

where we again used L'Hôpital's rule. Consequently, the derivative of the free energy at $\lambda = 0$ is

$$\frac{\mathrm{d}F}{\mathrm{d}\lambda}\bigg|_{\lambda=0} = \frac{1}{Z}\frac{\mathrm{d}Z}{\mathrm{d}\lambda}\bigg|_{\lambda=0} = \frac{-2b'\mu'}{2b'} = -\mu'.$$

The derivative for $\lambda \neq 0$ is straightforward to evaluate, and we find

$$\frac{\mathrm{d}F}{\mathrm{d}\lambda}\bigg|_{\lambda\neq 0} = \frac{(b'-\mu')e^{\lambda(b'-\mu')}-(-b'-\mu')e^{\lambda(-b'-\mu')}}{e^{\lambda(b'-\mu')}-e^{\lambda(-b'-\mu')}} - \frac{1}{\lambda} = -\mu' + b'\frac{e^{\lambda b'}+e^{-\lambda b'}}{e^{\lambda b'}-e^{-\lambda b'}} - \frac{1}{\lambda}.$$

Therefore, the final formula is

$$\frac{\mathrm{d}F(\lambda\,|\,b',\mu')}{\mathrm{d}\lambda} = \begin{cases} -\mu' & \lambda = 0 \\ -\mu' + b'\frac{e^{\lambda b'}+e^{-\lambda b'}}{e^{\lambda b'}-e^{-\lambda b'}} - \frac{1}{\lambda} & \lambda \neq 0 \end{cases}.$$

We determine $\lambda^*$ by setting this expression equal to zero.

Most simply, if $\mu' = 0$, then $\lambda^* = 0$ minimizes the free energy, and the MED is a uniform distribution. This result is intuitive. Without an explicit mean constraint, the zeroth-order MED with a finite contrast bound is a uniform distribution with zero mean, and the requirement that the distribution have zero mean is already satisfied.

If $\mu' \neq 0$, we need to solve

$$0 = -\mu' + b'\frac{e^{\lambda^* b'}+e^{-\lambda^* b'}}{e^{\lambda^* b'}-e^{-\lambda^* b'}} - \frac{1}{\lambda^*}.$$

We rearrange this expression to find

$$\frac{\mu'}{b'} = \frac{e^{\lambda^* b'}+e^{-\lambda^* b'}}{e^{\lambda^* b'}-e^{-\lambda^* b'}} - \frac{1}{\lambda^* b'} = \coth(\lambda^* b') - \frac{1}{\lambda^* b'},$$

where $\coth(x)$ is the hyperbolic cotangent function. If we define $f(x) = \coth(x) - \frac{1}{x}$, then we can rewrite the above equation as

$$\frac{\mu'}{b'} = f(\lambda^* b'),$$

and we get

$$\lambda^* = \frac{1}{b'}f^{-1}\left(\frac{\mu'}{b'}\right).$$

From the above equation, we observe that $\lambda^*$ depends only on $\frac{\mu'}{b'}$ and $\frac{1}{b'}$. From **Appendix 2—figure 1A**, we see that $\lambda^*$ monotonically increases with $\frac{\mu'}{b'}$. Intuitively, $\frac{\mu'}{b'}$ sets the relative scale of $\mu'$ in the contrast range and the degree of asymmetry in the distribution. For example, when $\mu'$ is zero, we know that $\lambda^*$ is zero and the MED is perfectly symmetric. As $\mu'$ deviates from 0, $\lambda^*$

deviates from zero and the distribution distorts asymmetrically from the uniform distribution to satisfy the non-zero mean. As the ratio of $\mu'$ and $b'$ grows larger, $\lambda^*$ has to deviate more from 0, and the distribution becomes increasingly asymmetric. When $\frac{\mu'}{b'}$ is small, $\lambda^*$ is roughly linear in $\frac{\mu'}{b'}$. Also note that there is a simple proportionality between $\lambda^*$ and $\frac{1}{b'}$. This dependence is easily understood by the requirement that $\lambda^* x$ is dimensionless.

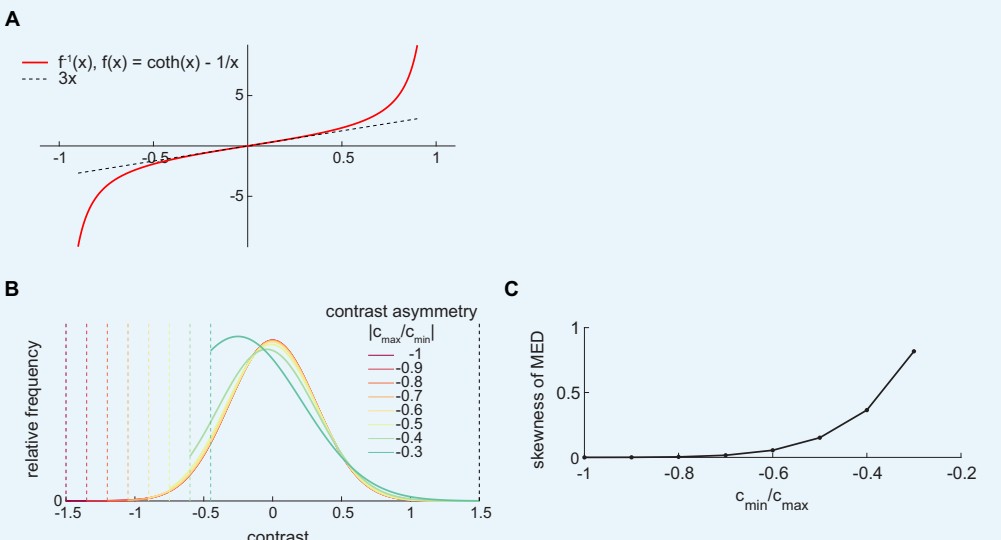

**Appendix 2—figure 1.** The shape of MED depends on the contrast range. (**A**) The shape of function $f^{-1}(x)$, where $f(x) = \coth(x) - 1/x$. (**B**) The mean-variance-constrained MED depends on the contrast range. The mean is constrained to be 0. The upper bound of the contrast range is fixed at 1.5, and the ratio of the lower-bound to upper bound changes from $-1$ to $-0.2$. As the contrast range becomes more asymmetric, the MED becomes more asymmetric. (**C**) When the contrast range is asymmetric, the mean-variance-constrained MED has induced non-zero skewness.

The mean-and-variance-constrained MED also depends on the finite contrast range. To simplify the discussion, we again set the zero point of x-axis such that the contrast range is symmetric around 0. Thus, our task is to find $\lambda_1, \lambda_2$ that minimize the free energy function,

$$\lambda^* = \left(\lambda_1^*, \lambda_2^*\right) = \underset{\lambda}{\operatorname{argmin}}\, F(\lambda_1, \lambda_2 | \mu_1, \mu_2, b) = \underset{\lambda}{\operatorname{argmin}}\, \log \int_{-b}^{b} dx \exp\left(\lambda_1(x - \mu_1) + \lambda_2(x^2 - \mu_2)\right),$$

where $\mu_1$ and $\mu_2$ are the shifted first-order and second-order moments imposed on the MED, and $[-b, b]$ is the shifted contrast range. As was the case for the first-order MED, when $\mu_1 = 0$, the distribution is symmetric and $\lambda_1^* = 0$. Furthermore, as the contrast bound becomes large, the distribution approaches the Gaussian distribution. However, when $\mu_1 \neq 0$, the distribution has to distort asymmetrically to achieve the non-zero mean, with large distortions occurring when the mean is within a few standard deviations of the boundary (**Appendix 2—figure 1B**). As a result, even though we did not require the distribution to have a specific skewness, the second-order MED can have non-zero skewness (**Appendix 2—figure 1C**).

These mathematical considerations lead to the design of our numerical methods. Operationally, for each individual image, we found its contrast range, $[c_{\min}, c_{\max}]$, and the largest contrast magnitude, $\delta c = \max(|c_{\min}|, |c_{\max}|)$. Then we constructed a symmetric contrast range around the constrained mean $\mu_1$, and computed the MED on $[\mu_1 - \delta c, \mu_1 + \delta c]$. This avoided inducing non-zero skewness in the mean-and-variance-constrained MED. For consistency, we used the same symmetric contrast range for the mean-variance-and-skewness-constrained MED. Since the contrast bound implicitly influences the shape of the MED and different individual images have different contrast ranges, this method takes guidance from natural scenes and uses *heterogenous* contrast ranges that match the contrast range of each

original natural scene. However, this heterogeneity introduces variability across images that could potentially impact the estimation accuracy we observed. Therefore, we also simulated secondary datasets where we used a common symmetric bound for all synthetic images. We chose this bound such that we could successfully solve the MED for the mean, variance, and skewness levels of most natural images.

