## [Decision Letter]

Thank you for submitting your article "Measured perceptual nonlinearities show how ON-OFF asymmetric processing improves motion estimation in natural scenes" for consideration by *eLife*. Your article has been reviewed by three peer reviewers, one of whom is a member of our Board of Reviewing Editors, and the evaluation has been overseen by Ronald Calabrese as the Senior Editor. The following individual involved in review of your submission has agreed to reveal their identity: Fred Rieke (Reviewer #2).

The reviewers have discussed the reviews with one another and the Reviewing Editor has drafted this decision to help you prepare a revised submission.

Summary:

This paper takes a close look into the natural scene statistics that influence behavior. In particular, the work investigates the importance of second- and third-order stimulus correlations on motion sensing in flies. Second-order correlations have been emphasized in past work, and a few past studies have suggested that third-order correlations also contribute, but this work provides the first comprehensive treatment of the issue. The paper presents an analysis of fly behavioral responses to moving patterns based on rigid rotations of static natural scenes, either with fully natural statistics, or modified parametric stimuli that retain the skewness in the contrast distribution of natural scenes. Analysis of behavioral responses to these stimulus variants indicates that the fly uses third -order computations effectively to counter some of the systematic (and well-known) variations introduced by the second-order computations prescribed by canonical motion detection models. Both the experiments and the analysis are presented clearly, and the authors make a convincing case for their conclusion that the third-order term improves velocity estimation in natural scenes. The computational strategies discovered here likely extend to other visual systems. The stimulus construction algorithm developed here could also be used for other model systems and analyses.

Essential revisions:

1) As pointed out in the Discussion section, the Volterra kernel approximation represents a general and systematic approach to the identification of nonlinear systems (with practical caveats, as also mentioned). But there is also a more fundamental issue: Volterra kernels are essentially high dimensional polynomial approximations, and as such, although they are general, they are extremely inefficient in describing compressive nonlinearities (and biological systems are essentially always compressive in their typical range of operation). Motion estimation by correlation is a second-order Volterra operation, and therefore diverges quadratically. To counter that divergence, the system must somehow normalize the response to the contrast variance, such as in contrast normalization. And this is what seems to happen in biological motion detectors: It is well known that motion responses saturate as a function of contrast.

In this light, the finding that the third-order Volterra kernel improves velocity estimates can then be simply seen as a corollary of the fact that the third-order kernel is essentially a "fitting parameter" which is necessarily better than a fit to second-order. Alternatively, as noted in the paper, the third-order behavior could be due to more or less fixed asymmetries in processing, or even due to nonlinear behavior in early vision. Something along these lines was proposed by Bulthoff and Goetz (Bulthoff and Goetz, 1979) to explain a motion illusion in human and fly vision (and that should be cited in this paper).

These points should be addressed fully in the text of the paper, both in the Results and in the Discussion.

2) Response predictions and quantification. The kernel approach estimates behavioral responses, but the predictions are then compared directly to image velocity. Implicit in this (at least not clearly stated in the text of the manuscript) is that the behavior is linearly related to the image velocity – or at least that the two can be swapped in the analysis. It would be good to clarify this issue somewhere around the second paragraph of the subsection “Third-order kernel improve velocity estimation in moving natural scenes” where this analysis is described. At present, the text could be misread to indicate that image velocity is being estimated directly.

3) In the Discussion section:

- The interpretation of the main result (that the third-order term improves velocity estimation in natural scenes) feels a bit narrow; the authors should remedy that in the Discussion section.

- Ties to neural data need to be made a bit more extensive and detailed. Is there evidence to suggest that the kinds of optimal temporal filters derived here are found in the fly brain? Perhaps suggest a few other places to look, beyond what's already discussed (measuring T4/T5 kernels).

- Comment a bit more on the limitations/advantages of considering only rigid rotations applied to static natural or naturalistic scene stats. Perhaps also comment on using walking versus flight, and note that flies spend most of their time walking.

- Comment a bit more on the potential behavioral relevance of the magnitude of the effect of the improvement from third-order filter input to second-order processing.

- How prevalent is positive skewness in different natural scenes? Are there scenes/environments where one should expect more or less skewness? Would visual systems that operate in specialized niches be expected to have different nonlinearities in their motion processing as a result, or will all visual systems require some amount of compensation in their second-order motion detectors?

Title revisions:

Replace "perceptual" with "behavioral"

Suggested stylistic revisions:

The papers dives into contrast-polarity sensitivity right away, before sufficiently motivating the research approach as a whole. It also seems like this feature might require a more intuitive figure to explain the effect to the uninitiated. Perhaps a few more words about the behavioral advantages of this level of careful motion dissection would be useful.

In the last paragraph of the Introduction, it would be good to broaden the perspective a bit on how filters are usually computed in behavioral and neural recording experiments, how this relates to other filter-finding techniques, and what the third-order kernels might reveal. For a broad audience, it might also be good to remind the reader what a Volterra expansion is and add a note about adding kernels to the expansion versus static nonlinearities, along the lines discussed above.

Here and there the paper reads a bit too much like a travel journal – a few examples:

- The paragraphs in the subsection “The structure of natural scenes induces noise in second-order motion estimates” all have something like "We did something…Then we observed something, etc."

- The legend to Figure 3 is full of "we did this or that".

Consider revising the text to make the language just a bit more formal and deliberate in guiding the reader through the major scientific findings in the paper, by motivating and previewing the steps taken in this careful and thorough work.

There are a number of typos that need to be corrected, through a close reading of the text.

---

## [Author Response]

Essential revisions:1) As pointed out in the Discussion section, the Volterra kernel approximation represents a general and systematic approach to the identification of nonlinear systems (with practical caveats, as also mentioned). But there is also a more fundamental issue: Volterra kernels are essentially high dimensional polynomial approximations, and as such, although they are general, they are extremely inefficient in describing compressive nonlinearities (and biological systems are essentially always compressive in their typical range of operation). Motion estimation by correlation is a second-order Volterra operation, and therefore diverges quadratically. To counter that divergence, the system must somehow normalize the response to the contrast variance, such as in contrast normalization. And this is what seems to happen in biological motion detectors: It is well known that motion responses saturate as a function of contrast.

We agree that a major problem of quadratic motion estimators is that they are too responsive to large contrast stimuli. We also agree that contrast normalization is one popular approach to ameliorate this issue. We have added a citation to Simoncelli and Heeger to point out how contrast normalization has to been applied to visual motion processing.

Our current approach leverages a simple algorithmic description to summarize multiple incompletely known nonlinear processing mechanisms, and remains agnostic to their origins in the circuit. The fly community is currently dissecting the biophysics of visual circuits in significant detail, and it is now clear that many distinct nonlinear mechanisms contribute to visual motion computation. As such, we agree that we’re studying phenomena that could conceivably relate to contrast normalization, but we think it is premature to conclude that contrast normalization is the origin of the effects we observe here.

Furthermore, typical contrast normalization methods generate contrast-*symmetric* nonlinear processing, for instance by normalizing by the variance of the inputs or by passing the (symmetric) motion signal itself through a normalizing or saturating process. In comparison to these forms of contrast-symmetric processing, the third-order kernel captured the contrast-*asymmetric* nonlinearity in the circuit, which is qualitatively different from standard contrast normalization.

We have addressed these issues through several changes in the manuscript. We made three changes to the Results section where the Volterra series expansion first appeared. First, we explicitly made the connection between the Volterra series and a polynomial expansion.

“Similar to the Taylor series from calculus, the Volterra series is a polynomial description of a nonlinearity, with a first-order kernel that describes linear transformations, a second-order kernel that captures quadratic terms, and higher-order kernels that combine to represent a wide variety of nonlinearities beyond the second-order.”

We then mention that Volterra series might converge inefficiently to the true mechanistically accurate nonlinearity and mentioned that there are convenient alternatives to our approach. Updated text:

“However, many polynomial terms can be needed to describe some nonlinearities. For instance, the polynomial description of a compressive, saturating nonlinearity is inefficient, and it can be easier to describe such transformations using alternative nonlinear model architectures, such as linear-nonlinear cascade models (Dayan et al., 2001).”

Finally, we pointed out that this expansion should be taken at the algorithmic, rather than the mechanistic level:

“We emphasize that the Volterra kernel description is explicitly algorithmic, as it aims to summarize the overall system processing without considering the mechanisms leading to this processing.”

This latter point permitted us to get into more details in the Discussion section. In particular, we added asymmetric contrast adaptation in the ON-OFF pathway as a potential nonlinearity that could lead to light-dark asymmetric motion processing. The text there was updated as follows:

“Yet it remains unclear whether asymmetric responses of T4 and T5 are inherited from upstream neurons. For instance, contrast adaptation could differ between the two pathways (Chichilnisky and Kalmar, 2002).”

In this light, the finding that the third-order Volterra kernel improves velocity estimates can then be simply seen as a corollary of the fact that the third-order kernel is essentially a "fitting parameter" which is necessarily better than a fit to second-order.

We do not think that this assertion is correct. It is true that one will better approximate a system by adding more fitting parameters, and we indeed better predicted optomotor turning responses with the added third-order kernel. However, we fitted a model to explain optomotor turning behavior, but we evaluated the model’s performance on a different task, i.e., velocity estimation. There is no guarantee that the system responsible for optomotor behavior is also optimized for velocity estimation, nor that including a third-order Volterra term in the fit would improve velocity estimates in natural scenes.

When we evaluated the model fitted to optomotor behavior with the task of velocity estimation in natural scenes, we had three implicit hypotheses. 1. We hypothesize that the visual system is optimized for velocity estimation. 2. We hypothesize that the turning response is a readout of that velocity estimate. 3. We hypothesize that the system is tailored specifically to the idiosyncrasies of natural scenes, rather than working for arbitrary image ensembles. The results are consistent with our hypotheses, and our finding that the third-order fit does improve natural scene performance is an exciting non-trivial result of this paper. It is not a corollary to fitting a model with more parameters.

We first added an introductory Figure 3 paragraph in the Results section to explicitly state our assumption that optomotor behavior is a proxy for the fly’s internal estimate of the image velocity:

“The kernels were fit to turning behavior, so the output of the model to moving visual stimuli is the predicted optomotor turning response. […] Using the fitted behavioral model, we could thus investigate how accurately the fly’s velocity estimate tracks the true image velocity.”

We then elaborated our results of improving velocity estimation with the third-order kernel in the Discussion section:

“In this study, we first fit a Volterra series expansion to model the fly’s turning behavior in response to binary stochastic stimuli, and both second- and third-order terms in the Volterra series contributed to the turning behavior. […] Therefore, these results can be taken together to motivate the hypothesis that the magnitude of the fly’s turning response is determined by an internal estimate of velocity.”

Alternatively, as noted in the paper, the third-order behavior could be due to more or less fixed asymmetries in processing, or even due to nonlinear behavior in early vision. Something along these lines was proposed by Bulthoff and Goetz (Bulthoff and Goetz, 1979) to explain a motion illusion in human and fly vision (and that should be cited in this paper).

We agree that it’s interesting to ask how nonlinear circuits could develop sensitivity to third-order motion cues. In previous work, we have investigated a variety of circuit nonlinearities that could give rise to third-order motion processing (Fitzgerald and Clark, 2015), including contrast asymmetric front-end nonlinearities of the type suggested by Bulthoff and Goetz. Thank you for pointing us to this reference. We have now cited it. Our text around this issue now reads:

“Previous work has suggested that front-end nonlinearities could account for certain optomotor illusions in flies (Bülthoff and Götz, 1979), and it is conceivable that such nonlinearities could generate contrast asymmetric motion responses (Clark and Demb, 2016; Fitzgerald et al., 2011). […] Alternatively, nonlinear processing at the level of direction-selective T4 and T5 neurons could also generate the asymmetries we observed here.”

These points should be addressed fully in the text of the paper, both in the Results and in the Discussion.

We hope that you will find that these modifications have improved the paper and have fully addressed the concerns of the reviewers.

2) Response predictions and quantification. The kernel approach estimates behavioral responses, but the predictions are then compared directly to image velocity. Implicit in this (at least not clearly stated in the text of the manuscript) is that the behavior is linearly related to the image velocity – or at least that the two can be swapped in the analysis. It would be good to clarify this issue somewhere around the second paragraph of the subsection “Third-order kernel improve velocity estimation in moving natural scenes” where this analysis is described. At present, the text could be misread to indicate that image velocity is being estimated directly.

Thank you for pointing out that this issue was unclear in our manuscript. First, in line with comment 1, we extended the introduction to Figure 3 to state explicitly that our model fitted behavioral responses but predicted image velocity. Second, we have added a few sentences to clarify the role of the linear assumption in our chosen accuracy metric. In particular, before presenting the results of Figure 3C, we now say:

“We quantified how well the model’s responses predicted the image velocity using the Pearson correlation coefficient. This metric supposes that the model response and image velocity are linearly related, and its value summarizes intuitively the mean-squared-error of the best linear fit between the model’s output and the image velocity.”

3) In the Discussion section:- The interpretation of the main result (that the third-order term improves velocity estimation in natural scenes) feels a bit narrow; the authors should remedy that in the Discussion section.

Thank you for these suggestions. To address this comment, we first added a summary Discussion paragraph to broaden our interpretation of the main result:

“In this study, we first fit a Volterra series expansion to model the fly’s turning behavior in response to binary stochastic stimuli, and both second- and third-order terms in the Volterra series contributed to the turning behavior. […] Since skewed scenes are prevalent across natural environments, and many visual systems exhibit ON-OFF asymmetric visual processing (Chichilnisky and Kalmar, 2002; Jin et al., 2011; Mazade et al., 2019; Pandarinath et al., 2010; Ratliff et al., 2010; Zaghloul et al., 2003), many animals are likely to use similar strategies for motion perception.”

We also related our work to other fields. We first discussed how the methodology of Volterra series expansion is used in other fields of neuroscience:

“Polynomial approximations to complex nonlinear systems have also been useful in other domains of neuroscience. […] These motifs might relate to measurable properties of the neocortex (Song et al., 2005).”

We then discussed how ON-OFF asymmetric processing in other visual systems is connected to natural scenes statistics:

“This resonates with previous work showing that many visual systems process ON and OFF signals asymmetrically to improve other aspects of visual processing (Kremkow et al., 2014; Mazade et al., 2019; Pandarinath et al., 2010; Ratliff et al., 2010). […] ON-OFF asymmetric visual processing also varies in other ways, and there is evidence that contrast adaptation in ON and OFF pathways is different between primate and salamander retinas (Chander and Chichilnisky, 2001).”

In the end, we now discuss why modeling behavior at algorithmic level is beneficial:

“David Marr famously asserted that neural computation needs to be understood at both the algorithmic and implementational levels (Marr and Poggio, 1976). […] Instead of algorithm and mechanism providing parallel or hierarchical goals, they should be treated as parts of one integrated understanding of the circuit.”

- Ties to neural data need to be made a bit more extensive and detailed. Is there evidence to suggest that the kinds of optimal temporal filters derived here are found in the fly brain? Perhaps suggest a few other places to look, beyond what's already discussed (measuring T4/T5 kernels).

To tie the measured behavioral kernel to neural data, we have added a Discussion section on potential mechanisms for the measured asymmetries. We first point out that asymmetric front-end filtering could, but would not necessarily, result in the measured third-order kernel, then discuss how asymmetric processing could originate upstream of T4/T5, in T4/T5, or differential weighting of T4 and T5 outputs downstream.

“Indeed, differentially affecting T4 and T5 activity, either through direct silencing or by manipulating upstream neurons, alters the behavioral responses of flies to triplet correlations (Clark et al., 2014; Leonhardt et al., 2016), and parallel experiments in humans similarly find that contrast-asymmetric responses are mediated by neurons separately modulated by moving ON and OFF edges (Clark et al., 2014). […] By measuring behavior and distilling the abstract algorithmic properties of the system, we will be able to constrain the contributions of individual circuit components without confining ourselves to an overly narrow class of mechanistic models.”

- Comment a bit more on the limitations/advantages of considering only rigid rotations applied to static natural or naturalistic scene stats. Perhaps also comment on using walking versus flight, and note that flies spend most of their time walking.

Thank you for this suggestion. We now comment on the limitations and advantages of these rigid rotations when we introduce them in Figure 1:

“This rigid translation of images mimics the motion produced by an animal’s pure rotation, during which visual objects all move at the same rotational velocity and occlusion does not change over time. Real motion through an environment generates more complex signals than this, but rigid translations are straightforward to compute and rotational visual stimuli are known to induce the rotational optomotor response that we focus on in this manuscript.”

We also note at the outset that *Drosophila* spends a large fraction of its life walking:

“Flies spend a large portion of their lives standing and walking on surfaces, making walking optomotor responses ethologically critical (Carey et al., 2006).”

- Comment a bit more on the potential behavioral relevance of the magnitude of the effect of the improvement from third-order filter input to second-order processing.

We added a Discussion section to address this suggestion. We started by comparing the behavioral magnitude in third-order glider stimuli and the computational benefits of the third-order kernel, we then compared the magnitude of the improvement added by the third-order kernel with other mechanisms such as spatial averaging, and we finally pointed out that the third-order kernel may underestimate the full improvement provided by its underlying mechanisms. This new section reads as follows:

“The HRC has explained a large number of phenomena and neural responses, and it is reasonable to ask how much we’ve gained by extending its second-order algorithmic description of fly behavior to a third-order one. […] It will be interesting to investigate whether mechanistically accurate models that explain the origin of the third-order kernel also reveal larger performance improvements.”

- How prevalent is positive skewness in different natural scenes? Are there scenes/environments where one should expect more or less skewness? Would visual systems that operate in specialized niches be expected to have different nonlinearities in their motion processing as a result, or will all visual systems require some amount of compensation in their second-order motion detectors?

Thank you for these interesting questions. There are studies showing that contrast adaptation in ON and OFF pathways in primates differs from those pathways in salamanders; that could be the result of different visual environments. We now added a section that discusses natural scene statistics in different habitats, and how natural environments would combine with early processing to shape the strategy of motion computation. This new section reads as follows:

“Adaptations to differences in both habitat and early sensory processing could potentially explain these divergences. […] These possibilities are not mutually exclusive, and in both cases, the early visual processing must work in concert with the downstream motion detectors to form robust and consistent perceptions.”

Title revisions:Replace "perceptual" with "behavioral"

Thank you for this suggestion. We had many discussions about how well ‘perceptual’ fit here. We have now revised our title to read: “Asymmetric ON-OFF processing of visual motion cancels variability induced by the structure of natural scenes”.

Suggested stylistic revisions:The papers dives into contrast-polarity sensitivity right away, before sufficiently motivating the research approach as a whole. It also seems like this feature might require a more intuitive figure to explain the effect to the uninitiated. Perhaps a few more words about the behavioral advantages of this level of careful motion dissection would be useful.

Thank you for this suggestion. We now spend time in the Introduction motivating the systems analysis and studying motion computation in terms of spatiotemporal correlations before we introduce contrast-polarity sensitivity:

“Higher-order correlations could also contribute to motion computation, and Bayes optimal visual motion estimators can be written as a sum of terms specialized for detecting different correlation types (Poggio and Reichardt, 1973; Potters and Bialek, 1994).[…] This sensitivity to triplet spatiotemporal correlations shows that motion perception incorporates cues neglected by canonical motion detectors.”

In the last paragraph of the Introduction, it would be good to broaden the perspective a bit on how filters are usually computed in behavioral and neural recording experiments, how this relates to other filter-finding techniques, and what the third-order kernels might reveal. For a broad audience, it might also be good to remind the reader what a Volterra expansion is and add a note about adding kernels to the expansion versus static nonlinearities, along the lines discussed above.

Thank you for this suggestion. In line with previous comments, we now added a brief introduction to the Volterra expansions in the Introduction Section. In line with Essential Revisions 1, in the Results section, we added more detailed explanation for Volterra series expansion, and relate it to other modeling techniques for nonlinear system. We also added prose to motivate why we choose Volterra expansion in the study and what the third-order kernel might reveal:

“Volterra kernels are useful for studying visual motion processing because they allow us to rigorously group response properties by their order (Fitzgerald et al., 2011; Potters and Bialek, 1994), thereby permitting us to clearly describe both canonical and contrast polarity-dependent components of the behavior. […] The third-order kernel directly measures sensitivities to triplet spatiotemporal correlations and probes ON/OFF asymmetries in motion processing.”

Here and there the paper reads a bit too much like a travel journal – a few examples:- The paragraphs in the subsection “The structure of natural scenes induces noise in second-order motion estimates” all have something like "We did something…Then we observed something, etc."

Thank you. We have revised to add motivation throughout.

- The legend to Figure 3 is full of "we did this or that".

The legend has been updated to avoid this. Thank you.

Consider revising the text to make the language just a bit more formal and deliberate in guiding the reader through the major scientific findings in the paper, by motivating and previewing the steps taken in this careful and thorough work.

We have revised the text throughout to eliminate some of this informality and to better motivate the analyses we performed.

There are a number of typos that need to be corrected, through a close reading of the text.

As we revised and re-read the manuscript, we have corrected typos we found. We hope we caught them all.